# The Omni-Expert: A Computationally Efficient Approach to Achieve a Mixture of Experts in a Single Expert Model

**Sohini Saha**[1]     **Mezisashe S. Ojuba**[2]     **Leslie M. Collins**[1]     **Boyla O. Mainsah** [1*]

[1]Duke University, Durham, NC, USA; [2]Howard University, Washington, DC, USA

## Abstract

Mixture-of-Experts (MoE) models have become popular in machine learning, boosting performance by partitioning tasks across multiple experts. However, the need for several experts often results in high computational costs, limiting their application on resource-constrained devices with stringent real-time requirements, such as cochlear implants (CIs). We introduce the Omni-Expert (OE) - a simple and efficient solution that leverages feature transformations to achieve the 'divide-and-conquer' functionality of a full MoE ensemble in a single expert model. We demonstrate the effectiveness of the OE using phoneme-specific time-frequency masking for speech dereverberation in a CI. Empirical results show that the OE delivers statistically significant improvements in objective intelligibility measures of CI vocoded speech at different levels of reverberation across various speech datasets at a much reduced computational cost relative to a counterpart MoE.

## 1   Introduction

Mixture-of-Experts (MoE) models [1, 2] have emerged as powerful and flexible architectures for machine learning (ML) tasks that require specialized fine-tuning of complex tasks, such as language modeling and computer vision. However, MoE models pose computational challenges, in terms of power consumption, processing capability, latency and memory capacity, as the number of experts increases. Hardware considerations are highly relevant for deployment in resource-constrained edge devices, especially for applications with real-time inference constraints, such as auditory prostheses. Thus, there is a need for lightweight adaptations of MoE models or alternative strategies that preserve performance while meeting resource limitations.

In this work, we focus on the cochlear implant (CI), a medical device that restores hearing to individuals with profound hearing loss by converting sound to electrical pulses that directly stimulate an impaired cochlea. Most CI users generally have good speech understanding in quiet conditions; however, CI users struggle to understand speech in challenging acoustic environments with noise and reverberation [3]. Some modern hearing aids now incorporate deep neural networks (DNNs) either for acoustic scene selection to (de)activate specific features, to control channel-specific gains for noise reduction and for separating noise and speech, enabled by more powerful on-chip processors, e.g., [4–6]. DNNs have been investigated mostly for speech denoising in CIs [7–11]; however, to our knowledge, no DNNs have been deployed in current commercially available CI sound processors. Unlike hearing aids that primarily function to amplify sound, CIs have a higher functional burden of acoustic-to-electric signal conversion, wireless signal transmission and electrical stimulation. In addition, real-time CI sound processing must be causal with time delays that are within tolerable limits of audiovisual asynchrony for CI users; preferably <10 ms [12].

---

*Corresponding author: boyla.mainsah@duke.edu

39th Conference on Neural Information Processing Systems (NeurIPS 2025).

Real-time speech enhancement models in CI systems must strike a balance between computational efficiency, performance and latency, making scaling with traditional MoE-based solutions potentially impractical. We propose a novel network architecture that eliminates the need for multiple experts while retaining the specialization benefits of MoE models. Our contributions are as follows:

- We introduce the *Omni-Expert* (OE), a computationally efficient alternative to MoE. The OE model achieves the effectiveness of a full MoE ensemble in a single network by using subtask-specific transformations to partition the feature space into distinct regions that correspond to a specialized expertise. This allows a single expert network to maintain subtask specialization while operating with much reduced computational overhead.

- We demonstrate the effectiveness of the OE in achieving superior performance relative to a counterpart MoE in a task of phoneme-based speech dereverberation in CIs.

- We conduct additional experiments to analyze the effect of feature transformation components on the performance of the OE model in the speech dereverberation task.

## 2   Related Work

**Speech Enhancement in Cochlear Implants.** Currently, CIs incorporate several signal processing solutions for noise management, such as beam-forming and signal-to-noise ratio (SNR)-based noise reduction [13–15]; however, there is no solution that directly addresses reverberation. Even in the absence of noise, individuals with auditory prostheses often struggle to understand speech in reverberation [16], which can negatively impact their quality of life; for example, learning experiences [17, 18] typically involve single talker scenarios in a classroom/lecture hall. Strategies for noise reduction are not as effective in fully addressing reverberation due to differences in how noise and reverberation distort speech signals. Noise distortions are additive and do not depend on the target speech, while reverberant distortions are delayed and attenuated copies of the target speech.

A common approach for speech enhancement is *time-frequency* (T-F) *masking*, where a gain matrix (or mask) is applied to a T-F representation of the degraded speech to separate segments dominated by speech and acoustic distortions. An ideal mask is calculated based on a measure of distortion of a degraded speech signal relative to its clean counterpart. A typical *ideal ratio mask* (IRM) with mask values ranging from 0 to 1 is computed according to [19]:

$$0 \leq M(t,f) = \left( \frac{|S(t,f)|^2}{|S(t,f)|^2 + |N(t,f)|^2} \right)^{0.5} \leq 1 \tag{1}$$

$$\hat{S}(t,f) = M(t,f) \cdot X(t,f) \tag{2}$$

where $M(t,f)$ represents the ratio mask; $S(t,f)$ and $X(t,f)$ represent the clean and degraded speech signals, respectively; $N(t,f)$ represents the noise; and $\hat{S}(t,f)$ represents the enhanced signal.

Studies have shown ideal T-F masks for dereverberation improve speech intelligibility for hearing-impaired listeners [20–25]. In real-world settings, the ideal mask needs to be estimated using only the reverberant speech signal. Traditional mask estimation algorithms for CI applications have relied on statistical features, such as kurtosis [25], linear prediction residuals [26], or estimated signal-to-distortion ratios [27], typically require room-specific tuning and do not generalize well across varying acoustic conditions. Research on speech enhancement in CIs now utilizes DNNs that offer better robustness and generalization across diverse acoustic conditions. However, improvements with advanced ML models come at the cost of increased computational complexity and latency.

A main challenge with mitigating reverberation is how to distinguish between wanted vs. unwanted speech with similar characteristics (i.e., reverberant reflections) from the same target speaker. In general, the performance of ML algorithms for speech dereverberation is frequency dependent. Mid-to high-frequency speech regions often include relatively long gaps in between phonemes, which allows late reverberant reflections to exponentially decay over time; thus, ML models can leverage features that capture this decaying pattern to better estimate and suppress unwanted reverberant speech. However, lower-frequency speech regions have more energy and relatively short gaps between phonemes, so late reverberant reflections are often interrupted by the next phoneme before

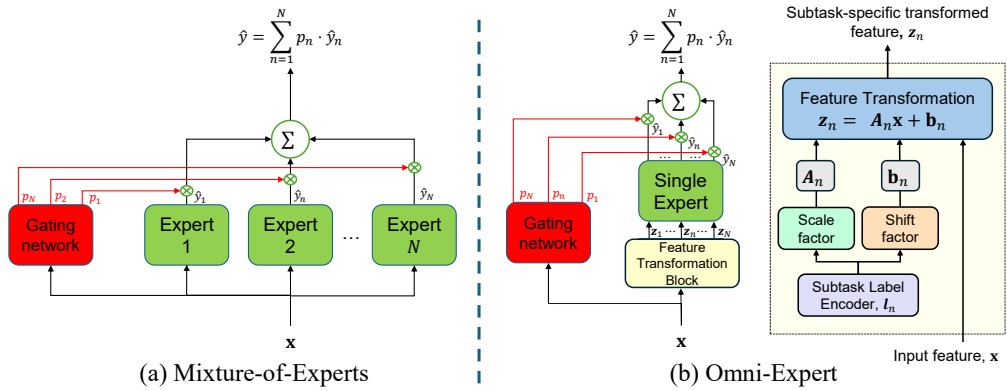

Figure 1: Illustration of (a) the Mixture-of-Experts (MoE) model and (b) our Omni-Expert (OE) model given an input feature, $\mathbf{x}$. The MoE model uses probabilities ($p_n$) from a gating network to weight outputs ($\hat{y}_n$) from *multiple* expert networks. In contrast, the OE model uses a *single* expert network and subtask-specific feature transformations ($\mathbf{z}_n$) to achieve the functionality of multiple experts. We train separate models to predict the scale and shift parameters for subtask-specific feature transformations and use a lookup table based on the subtask label during inference.

they can fully decay. Thus, it is more challenging to differentiate reverberant reflections from target speech in lower frequency speech regions because of more persistent, higher energy reverberation.

While speech signals exhibit high spectro-temporal variabilities, speech structure is generally predictable, and this predictability can be useful to inform speech enhancement. The energy of phonemes tends to be concentrated in specific frequency regions [28]. The presence of energy in other frequency regions during a phoneme utterance is likely indicative of an acoustic artifact; thus, knowledge of the current phoneme can be leveraged during mask estimation to better identify and remove acoustic distortions. Phoneme-based mask estimation models, where separate models are trained for different phonemes, have shown improved performance in speech denoising for automated speech processing applications [29–32]. Chu et al. [33] developed a phoneme-based mask estimation model for speech dereverberation in CIs based on a dense MoE (Figure 1a). Typically, speech enhancement models are applied as a preprocessing step prior to acoustic signal delivery to the CI. However, the model in [33] relies on causal features extracted from within the CI processing framework, a design that is feasible for real-time deployment in CIs.

**Mixture-of-Experts.** A key limitation of MoE approaches is the computational costs as the number of experts increases; in [33], the number of experts depends on the number of phonemes. In *sparse* MoE models [34–38], only a subset of experts are activated during inference. In [39, 40], the number of computations at inference in the MoE is reduced by *merging experts*, aggregating multiple expert parameters at inference time using gating weights. In sparse and merging of experts variants of the MoE model, multiple experts must still be trained, stored and maintained. To address this limitation, we have developed a novel technique that achieves multi-expert functionality in a single network.

**Conditional Computation.** Prior works [41, 42] have used conditional computation to improve model performance in multi-task problems. The MTFormer framework [41] employs a multi-task learning architecture consisting of a shared transformer-based feature extractor (encoder and decoder), followed task-specific branches for specialization. In conditional batch normalization [42], multilayer perceptrons are trained to learn additive adjustments to batch normalization scaling and shifting parameters of a pretrained convolutional neural network based on an external conditioning vector (e.g., a language embedding). Our approach applies conditional feature transformations directly to input features to achieve subtask specialization in a single network, and feature transformation parameters are learned jointly with the expert network.

## 3 The Omni-Expert (OE)

Our goal is to enable a single expert model to exhibit subtask-specific expertise based on the input features of the associated subtask. In sparse MoE models, a routing mechanism directs inputs to

the appropriate expert or set of experts. In contrast, our approach is to encode subtask selection for specialization *implicitly* in the feature space. To achieve this, we apply learned subtask-specific transformations that create homogenous features within a specific subtask and distinct features across subtasks. The OE model architecture, illustrated in Figure 1b, consists of three core components:

- A *feature transformation block* to apply subtask specific feature transformations based on the subtask label.
- A *single expert network*, which processes the transformed input features and performs the target task (in this case, mask estimation for speech dereverberation).
- A *gating/routing network*, which is used to weight the outputs produced by applying the single-expert network to the transformed features:

$$\hat{y} = \sum_{n=1:N} p_n(\mathbf{x}) E(\mathbf{z}_n) \tag{3}$$

where $\mathbf{x}$ is the input feature; $p_n(\mathbf{x})$, is the gating network probability of subtask $n$; $E(\mathbf{z}_n)$ is the subtask-specific output from the single expert model $E$ based on the transformed feature $\mathbf{z}_n$.

Instead of a simple linear transformation, an affine transformation offers greater flexibility by allowing a shift in the origin to better align the feature distribution for each subtask while preserving discriminability across subtasks. For a sparse affine transformation matrix, we restrict linear operations to scale transformations. The subtask-specific transformed feature $\mathbf{z_n}$ is defined as:

$$\mathbf{z}_n = \mathbf{A}_n \mathbf{x} + \mathbf{b}_n \tag{4}$$

where $\mathbf{x}$ is the input feature; $\mathbf{A}_n$ and $\mathbf{b}_n$ represent scale and shift transformations, respectively, for subtask $n$. For scale, $\mathbf{A}_n$ is a diagonal matrix, which simplifies to element-wise multiplication.

## 4 Methods

We demonstrate the efficiency of our OE over a MoE in the task of real-time speech dereverberation in CIs. The CI processing pipeline was implemented using the Nucleus MATLAB Toolbox [43]. All models were implemented in PyTorch [44] on an NVIDIA Titan V GPU.

### 4.1 Reverberation Model

The reverberant signal is modeled as the convolution of the clean, anechoic speech signal with a room impulse response (RIR). For proper time alignment with the delayed reverberant signal, the direct path component of the reverberant signal is used as the reference clean signal; see Appendix A.1.

### 4.2 Experimental Settings

**Datasets.** Speech utterances used for training were a randomly selected 8000-utterance subset (approximately 28 hours) of the 100-hour LibriSpeech corpus [45]. Recorded RIRs used for training were from the Brno University of Technology@FIT Reverberation Database [46]. Speech utterances used for testing were from speech datasets that are commonly used in listening studies: Hearing In Noise Test (HINT) [47]; and the City University of New York (CUNY) Male and Female datasets [48]. Recorded RIRs used for testing were selected from the Aachen Impulse Response database [49] to represent diverse rooms: office, stairway, lecture hall and church (Appendix A.1).

**Feature Extraction.** Causal T-F features were extracted from speech signals following the Advanced Combination Encoder (ACE) strategy based on the Nucleus CI system [50, 43]. The acoustic signal is segmented into 8 ms frames with a 2 ms overlap, processed via short-time Fourier transform yielding 65 frequency features, and log-compressed to reduce dynamic range. The log-compressed spectral features were used as inputs to both the phoneme classification and mask estimation models. Feature normalization was applied using the global mean and variance calculated from the training set.

**Phoneme Label Extraction.** The phoneme classes are 39 standard American English phonemes [28] and a nonphoneme class for silent gaps. Phoneme labels were generated from the direct path signal using forced alignment based on the LibriSpeech recipe in the Kaldi speech recognition toolkit [51]

and phoneme time stamps were converted into CI-based time units; example framewise phoneme labels are provided in Appendix A.2. A one-hot vector was used to encode phoneme labels.

**Model Architecture**. For real-time feasibility in CIs, we use lightweight recurrent neural networks. The models include: i) a single layer of 123 unidirectional long short-term memory (LSTM) units [52]; and ii) a single layer of 117 unidirectional gated recurrent units (GRU) [53] with a multi-head attention (A) layer [54]. The outputs of the GRU model following layer normalization are fed into the multi-head attention module (A), a residual connection around the attention block, followed by layer normalization (to stabilize gradients and accelerate convergence). The attention outputs are fused with the original GRU hidden state outputs via element-wise multiplication so that the final representation retains both the sequential encoding and the learned contextual weighting.

**Phoneme Classifier**. Each model is followed by a fully connected output layer with 40 sigmoidal units, corresponding to the 40 phoneme class labels [55]. Model parameters were randomly initialized from a uniform distribution with range [-0.1, 0.1]. Training was performed using stochastic gradient descent with a learning rate of 1e-5 and momentum of 0.9. Each training batch consisted of a single speech utterance split into 2-second non-overlapping segments to accommodate memory constraints and maintain sequence continuity. The model was optimized using cross-entropy loss. Training was terminated when the validation loss was unchanged for 10 consecutive epochs.

**Mask Estimation Models.** Models were trained to minimize the T-F signal loss function:

$$L = \frac{1}{TF} \sum_{t,f} \left( \hat{M}(t,f)X(t,f) - M(t,f)X(t,f) \right)^2 \tag{5}$$

where $X(t,f)$ represents the magnitude spectrum of the reverberant signal; $\hat{M}(t,f)$ represents the estimated mask; $M(t,f)$ represents the ideal mask; and $T$ and $F$ represent the number of time and frequency bins, respectively.

*Phoneme Independent.* Each model is followed by a fully connected layer, followed by an output layer with 65 sigmoidal units, representing the 65-dimensional IRM values [33]. The phoneme independent model was trained on the entire training dataset.

*Phoneme-based MoE.* The model consists of the phoneme classifier as the gating network and 40 expert networks for phoneme-specific mask estimation. Each expert network is structurally identical to the phoneme independent model but is trained only on data corresponding to its associated phoneme group. The phoneme classifier probabilities are used to weight the predictions of the phoneme-specific mask estimation models to get the final estimated mask.

*Phoneme-based OE.* The single expert model is structurally identical to the phoneme independent model. Input features are transformed based on a phoneme-specific transformation:

$$\boldsymbol{z}_n = \boldsymbol{a}_n \odot \boldsymbol{x} + \boldsymbol{b}_n \tag{6}$$

where $\boldsymbol{z}_n$ is the transformed feature vector; $\boldsymbol{x}$ is the input feature vector; $\boldsymbol{a}_n$ and $\boldsymbol{b}_n$ are the scale and shift factor vectors, respectively, for phoneme $n$; and $\odot$ represents element-wise multiplication.

The transformation parameters are predicted by two separate multilayer perceptrons (MLPs) with the one-hot phoneme encoding as input. Both MLPs have an input size of 40 (number of phoneme classes) and an output size of 65 (matching the input feature size). The scale MLP uses ReLU activation, while the shift MLP uses LeakyReLU to introduce nonlinearities [56]. The transformation parameters are precomputed after training and stored for all 40 phonemes, effectively reducing the process during inference to a lookup table.

*Model Training.* The phoneme independent model was initialized using weights drawn from a uniform [-0.1, 0.1] distribution [57]. The phoneme-specific mask estimation networks of the MoE and OE models were initialized using pre-trained weights from the phoneme independent model. Models were trained using the Adam optimizer [58] with a learning rate of 1e-3 and momentum coefficients of 0.9 and 0.999. Training batches consisted of 16 speech utterances segmented into 2 second chunks. Training was terminated when no improvement was observed for 10 consecutive epochs.

## 4.3   Performance Evaluation

CI vocoded speech was generated by acoustic signal resynthesis from CI electrodograms (CI electrical stimuli patterns) with a sine wave vocoder and used to predict speech intelligibility in CI users.

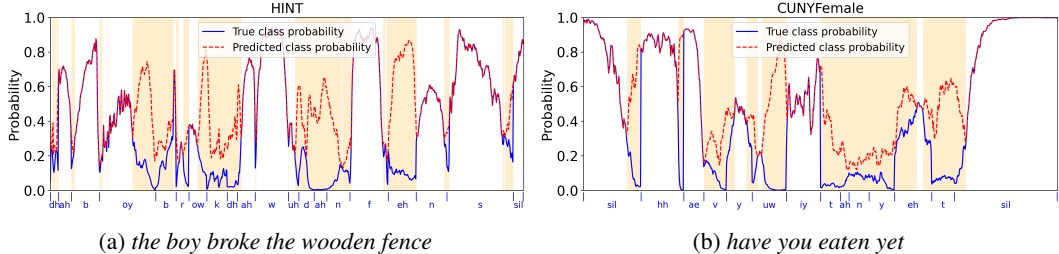

(a) *the boy broke the wooden fence*          (b) *have you eaten yet*

Figure 2: Framewise probabilities of true and predicted phoneme classes of reverberant speech utterances from (a) HINT and (b) CUNY-Female datasets using the GRU+A classifier. True class labels are annotated on the $x$-axis. Shaded regions indicate incorrect predictions.

**Objective Intelligibility Metrics.** Intelligibility of CI vocoded speech was predicted using objective metrics that have been shown to be predictive of speech intelligibility in CI listening studies [59]: the speech-to-reverberation modulation energy ratio for CI users (SRMR-CI) [60, 61]; and the short-time objective intelligibility (STOI) metric [62] with the direct path signal used as the reference signal [63]. The STOI metric was developed for evaluating speech intelligibility in noise and excludes frames with silent gaps in clean speech. Since reverberation occurs in silent speech gaps, the removal of reverberant reflections in the same silent speech gap will not be captured by the STOI metric. The SRMR metric was originally developed for evaluating speech intelligibility in reverberation for normal hearing listeners [60], and later adapted for CI users (SRMR-CI) by using the CI filterbanks [61]. Thus, the SRMR-CI metric provides a more reliable speech intelligibility predictor for CIs in reverberation. We include STOI for completeness and to facilitate comparison with prior literature.

**Benchmarks.** Chu et al. [33] showed improvements in SRMR-CI and STOI scores, as well as intelligibility of CI vocoded speech in normal hearing listeners, with a phoneme-based MoE model relative to a (conventional) phoneme independent mask estimation model. The objective here is to demonstrate that the OE model achieves at a minimum similar performance as the MoE model with much reduced complexity. Oracle conditions include mask estimation with ideal phoneme knowledge (i.e., perfect phoneme classification), the IRM and the direct path signal.

**Statistical Analysis.** Statistical tests were performed in R. A two-way repeated measures ANOVA was performed with within-utterance fixed factors of room and mask type, interaction between mask type and room, and a random factor of speech utterance. Mauchly's test [64] was used to assess the sphericity assumption; degrees of freedom were adjusted using the Greenhouse-Geisser correction where necessary. For statistically significant main effects or interactions, post-hoc pairwise comparisons were performed using estimated marginal means with Tukey's multiple comparisons correction. All tests were conducted at a significance level of 0.05.

## 5   Results

### 5.1   Phoneme Classification

Figure 2 show frame-wise phoneme classifier predictions using the GRU+A classifier applied to sample reverberant speech utterances. Frame-wise class balanced phoneme classification accuracies are summarized in Table 1. The phoneme distributions, classification confusion matrices and additional framewise classification results are provided in Appendix A.5.

Table 1: Class-Balanced Phoneme Classification Accuracies (%) Across Test Datasets

| Dataset | Long Short-term Memory | | | | Gated Recurrent Unit + Attention | | | |
|---|---|---|---|---|---|---|---|---|
| | Church | Office | Lecture | Stairway | Church | Office | Lecture | Stairway |
| CUNY-Female | 20.98 | 26.26 | 25.41 | 27.65 | 24.30 | 35.29 | 32.89 | 35.91 |
| CUNY-Male | 20.11 | 26.63 | 23.41 | 25.36 | 27.61 | 39.52 | 34.11 | 36.85 |
| HINT | 22.44 | 31.39 | 28.38 | 31.2 | 33.02 | 47.82 | 43.39 | 46.78 |

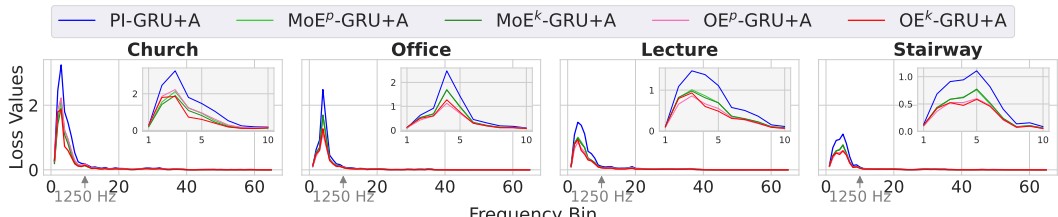

Figure 3: Average signal loss across frequency bins with phoneme independent mask estimation and mask estimation with mixture-of-experts (MoE) and Omni-expert (OE) models with predicted ($p$) and ideal ($k$) phoneme knowledge using gated recurrent unit + attention (GRU+A) models.

## 5.2 Mask Estimation

Figure 3 shows the mean signal loss across frequency bins to visualize the frequency-dependent impact of mask estimation. In general, signal loss is highest in lower frequency regions (< 1250 Hz), reflecting the difficulty in mitigating low frequency reverberation. Phonemes typically range from 70-200 ms [65, 66], making phoneme classification based on an 8 ms frame a hard task. Even at low accuracies (Table 1), predicted phoneme knowledge is still beneficial to mask estimation, with a progressive increase in performance from phoneme independent to MoE to OE models. Phonemes with similar time-frequency characteristics are likely to be confused with each other (Appendix A.5). The weighting of phoneme-specific masks reduces the impact of phoneme misclassifications. With known phonemes, the OE provides a higher performance upperbound than the MoE. This indicates that encoding subtask-specific cues via feature transformations is more effective for specialization vs. specialized experts with subtask partitioning of the original feature space.

## 5.3 Objective Speech Intelligibility

Sample electrodograms are shown in Figure 4a with annotations of target speech and (late) reverberant reflections; corresponding spectrograms are shown in Appendix A.3. Room-specific statistical results of SRMR-CI and STOI scores are shown in Figure 4b; summary statistics are provided in Appendix A.6. Aggregate statistical results are summarized in Table 2. Performance trends of objective speech intelligibility measures are generally consistent with those of mask estimation. The higher performance bound with the OE provides more robustness to the impact of phoneme prediction errors.

Table 2: Objective intelligibility scores (estimated marginal mean $\pm$ 95% confidence interval) for the reverberant signal (Rev), direct path signal (DP), and across different mask estimation methods: ideal ratio mask (IRM), phoneme independent model (PI), phoneme-based mask predicted by mixture-of-experts/Omni-Expert with ideal phoneme knowledge ($MoE^k$/ $OE^k$), and using phoneme classifier probabilities ($MoE^p$/ $OE^p$). Bold indicates the highest performance among the non-oracle models.

| Long Short-Term Memory (LSTM) | | | | | | | | |
|---|---|---|---|---|---|---|---|---|
| **Metric** | **Rev** | **PI** | **$MoE^p$** | **$MoE^k$** | **$OE^p$** | **$OE^k$** | **IRM** | **DP** |
| SRMR-CI | 1.302 $\pm$0.007 | 1.733 $\pm$0.009 | 1.744 $\pm$0.009 | 1.841 $\pm$0.009 | **1.794** $\pm$**0.010** | 1.938 $\pm$0.009 | 2.187 $\pm$0.008 | 2.447 $\pm$0.009 |
| STOI | 0.719 $\pm$0.001 | 0.797 $\pm$0.001 | **0.807** $\pm$**0.001** | 0.822 $\pm$0.001 | **0.807** $\pm$**0.001** | 0.836 $\pm$0.001 | 0.972 $\pm$0.000 | 1.000 $\pm$0.000 |
| **Gated Recurrent Unit+Attention (GRU+A)** | | | | | | | | |
| **Metric** | **Rev** | **PI** | **$MoE^p$** | **$MoE^k$** | **$OE^p$** | **$OE^k$** | **IRM** | **DP** |
| SRMR-CI | 1.302 $\pm$0.007 | 1.873 $\pm$0.011 | 1.948 $\pm$0.010 | 1.945 $\pm$0.009 | **2.014** $\pm$**0.011** | 2.113 $\pm$0.010 | 2.187 $\pm$0.008 | 2.447 $\pm$0.009 |
| STOI | 0.719 $\pm$0.001 | 0.812 $\pm$0.001 | **0.829** $\pm$**0.001** | 0.833 $\pm$0.001 | **0.829** $\pm$**0.001** | 0.850 $\pm$0.001 | 0.972 $\pm$0.000 | 1.000 $\pm$0.000 |

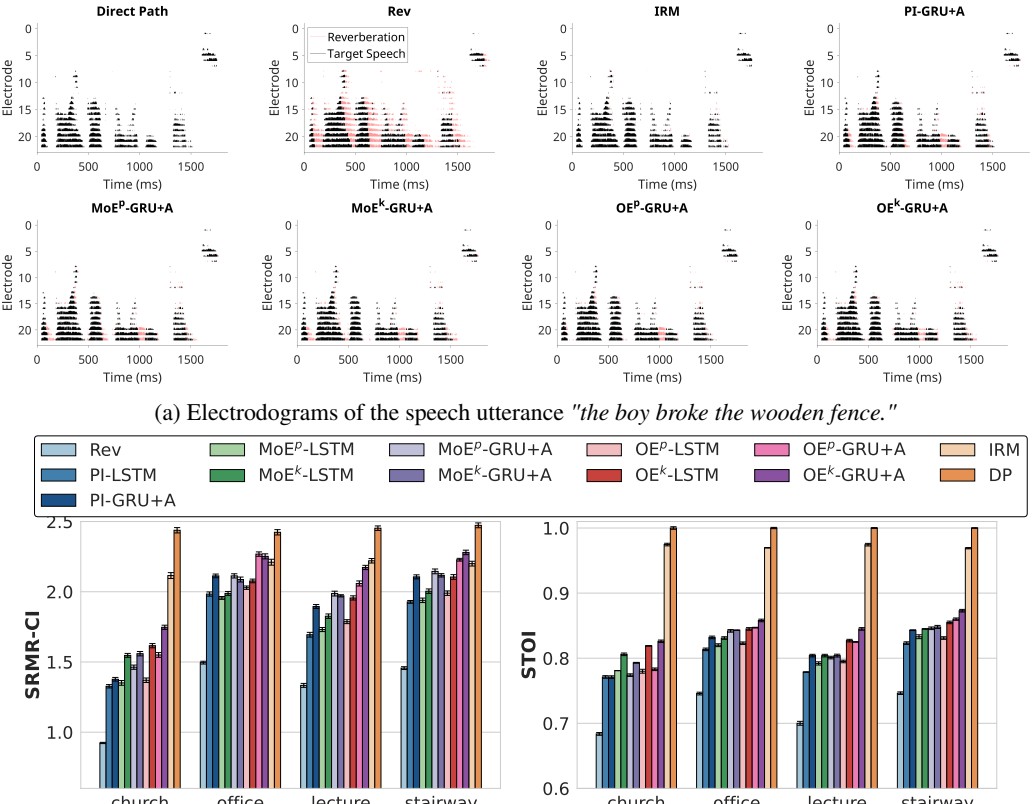

(a) Electrodograms of the speech utterance *"the boy broke the wooden fence."*

(b) Room-specific estimated marginal means and 95% confidence intervals of speech-to-reverberation modulation energy ratio for cochlear implant users (SRMR-CI) and short-time objective intelligibility (STOI) scores.

Figure 4: (a) Example electrodograms of a speech utterance and (b) room-specific statistical results of objective speech intelligibility measures of cochlear implant vocoded speech generated for direct path (DP), reverberant speech (Rev), enhanced reverberant speech after applying the ideal ratio mask (IRM) and estimated masks with the phoneme independent (PI) model, mixture-of-experts model with predicted and known phonemes ($MoE^{p/k}$) and Omni-Expert model with predicted and known phonemes ($OE^{p/k}$) for long short-term memory (LSTM) and gated recurrent unit + attention (GRU+A) networks.

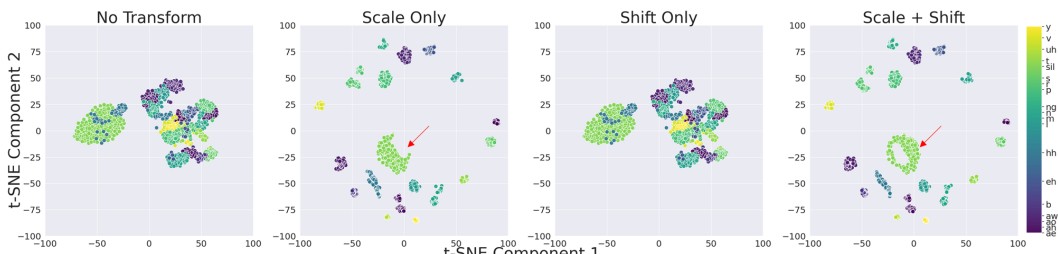

Figure 5: Visualization of phoneme-specific features from a subset of randomly selected phoneme frames (N = 1000) of reverberant speech from the CUNY Female speech dataset in the stairway room. Column panels represent features: before applying transformations; with scale-only; shift-only; and scale + shift transformations. Arrows indicate an example of a visually discernible impact of a shift transformation on a phoneme cluster. t-distributed stochastic neighbor embedding (t-SNE) was used.

## 5.4 Ablation Analysis

The rest of the paper presents aggregate results. Room-specific results are provided in the Appendix.

**Feature Transformation Type.** The contribution of each feature transformation was assessed with isolated (i.e., shift only or scale only) and combined (i.e., shift and scale) transformations. Figure 5 shows visualizations of features with respective transformations. The scale transformation enhances the separability of phoneme-specific feature clusters, while the shift transformations adjusts the feature offsets for better alignment. Aggregate statistical results are summarized in Table 3. Overall, applying scaling or shifting has a significant impact on the objective speech intelligibility metrics to a similar extent relative the non-transformed features. However, the combined transformation yields the highest improvements in SRMR-CI and STOI scores, Table 3.

Table 3: Objective intelligibility scores (estimated marginal mean $\pm$ 95% confidence interval) with the Omni-Expert model with predicted phonemes ($OE^p$) across different types of feature transformations, scale (Sc) only, shift (Sh) only, scale + shift (default) and no transformation. Bold indicates the highest performance. LSTM, long short-term memory; GRU+A, gated recurrent unit + attention.

| Metric | $OE^p$-LSTM | | | | $OE^p$-GRU+A | | | |
|---|---|---|---|---|---|---|---|---|
| | None | Sh Only | Sc Only | Sc + Sh | None | Sh Only | Sc Only | Sc + Sh |
| SRMR -CI | 1.683 $\pm$0.009 | 1.711 $\pm$0.010 | 1.706 $\pm$0.009 | **1.794** $\pm$**0.010** | 1.923 $\pm$0.011 | 1.987 $\pm$0.011 | 2.000 $\pm$0.011 | **2.014** $\pm$**0.011** |
| STOI | 0.793 $\pm$0.001 | 0.792 $\pm$0.001 | 0.793 $\pm$0.001 | **0.807** $\pm$**0.001** | 0.819 $\pm$0.001 | 0.826 $\pm$0.001 | **0.829** $\pm$**0.001** | 0.829 $\pm$**0.001** |

**Feature Transformation Position.** We also investigated the impact of applying the feature transformation at different layer positions during mask estimation: prior to the input of the model (default), after the hidden layer, and both the input and hidden layers. Aggregate results are summarized in Table 4. Overall, applying feature transformation at least at the input layer is more effective than applying the transformation only at the hidden layer.

Table 4: Objective intelligibility scores (estimated marginal mean $\pm$95% confidence interval) with the Omni-Expert model with predicted and known phonemes ($OE^{p/k}$) across different feature transformation positions: after the hidden layer (H), prior to the input to the model (I) (default) and both the input and hidden layers (I + H). Bold indicates the highest performance among the non-oracle models. LSTM, long short-term memory; GRU+A, gated recurrent unit + attention.

| Metric | $OE^p$-LSTM | | | $OE^k$-LSTM | | |
|---|---|---|---|---|---|---|
| | H | I | I + H | H | I | I + H |
| SRMR-CI | 1.764 $\pm$0.009 | 1.794 $\pm$0.010 | **1.805** $\pm$**0.010** | 1.863 $\pm$0.009 | 1.938 $\pm$0.009 | 1.947 $\pm$0.009 |
| STOI | 0.803 $\pm$0.001 | **0.807** $\pm$**0.001** | 0.805 $\pm$0.001 | 0.824 $\pm$0.001 | 0.836 $\pm$0.001 | 0.835 $\pm$0.001 |

| Metric | $OE^p$-GRU+A | | | $OE^k$-GRU+A | | |
|---|---|---|---|---|---|---|
| | H | I | I + H | H | I | I + H |
| SRMR-CI | 1.367 $\pm$0.008 | **2.014** $\pm$**0.011** | 2.004 $\pm$0.010 | 1.387 $\pm$0.006 | 2.113 $\pm$0.010 | 2.073 $\pm$ 0.010 |
| STOI | 0.693 $\pm$0.001 | **0.829** $\pm$**0.001** | 0.822 $\pm$0.001 | 0.621 $\pm$0.002 | 0.850 $\pm$0.001 | 0.842 $\pm$0.001 |

## 5.5 Model Complexity

Model size and training times are shown in Figure 6; detailed values are provided in Appendix A.4. The number of parameters and the computation load are obtained using the opensource *ptflops* package [67]. The OE model achieves comparable to superior performance with a much smaller model size and faster training time relative to the MoE model. Each expert in the MoE model is trained only on phoneme-specific data and the reduced amount of

training data per expert model results in a longer training time. In contrast, the OE model uses the full training dataset while still benefiting from sub-task specialization via the feature transformations. Note that the models for shift and scale factor estimation are only used during OE training; in this case, the mapping from phoneme label to feature transformation is deterministic, so only the subtask-specific transformation factors are needed during inference.

## 6 Conclusion

We introduced the Omni-Expert, a high-performing and compute-efficient alternative to achieving a mixture of expertise in the same model. By conditioning feature transformations on a subtask, the OE model learns to partition the input space into subtask-specific regions, effectively scaling the functionality of multiple experts without incurring the added computational costs. The current OE configuration assumes the subtasks are known to determine the number of experts, as is the case with phonemes. Alternative variants of the OE may be needed in other applications where the specialized subtasks are not as well-defined.

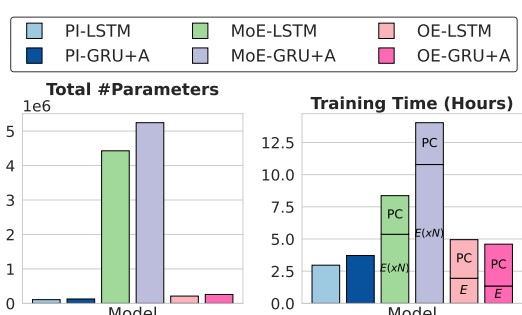

Figure 6: Complexity of phoneme independent (PI), mixture-of-experts (MoE, $N = 40$ experts) and Omni-Expert (OE) models using long short-term memory (LSTM) and gated recurrent unit + attention (GRU+A) networks for speech dereverberation in cochlear implants. MoE and OE model training times are marked by the phoneme classifier (PC, same for MoE and OE) and expert (E) networks. Training times based on a Titan V GPU.

While this work focused on speech dereverberation in cochlear implants, real-world listening scenarios will also include combinations of reverberation and ambient noise (e.g., multi-talker babble, equipment noise, etc). Results with the current mask estimation models trained *only* on reverberant speech and applied to speech in a variety of noisy reverberant settings are presented in Appendix A.8. As expected, there is a significant drop in performance across all models as the signal-to-noise ratio increases. The OE with known phonemes still outperforms the counterpart MoE, indicating that the learned subtask feature transformations are more robust to training/test data mismatch.

**Limitations and Future Work.** Performance trends observed with objective speech intelligibility measures may not necessarily translate to speech understanding in real-world settings; the test datasets may not fully represent the variety in speech patterns and reverberant room conditions. Additional performance improvements can be obtained with alternative OE model architectures, more diverse datasets for training and advanced training techniques. A relatively lightweight OE model for speech enhancement is more practical for CI deployment and improvements in CI processor chip technology are expected over time. Further work is needed to evaluate real-time feasibility and real-world utility in clinical studies with CI users.

**Societal Impact.** The OE provides a promising solution for developing compact, high-performing speech enhancement models to increase speech understanding for CI users in challenging listening environments, which can improve the quality of life of CI users. Real-time speech enhancement is also relevant for individuals with hearing aids and automatic speech processing applications that require minimal latency, such as live transcription. More broadly, the computational efficiency and scalability of the OE technique have the potential to address the computational bottleneck associated with MoE in other applications. The core architectural design of substituting multi-expert inference with subtask-specific transformations applied to a single expert is in principle domain-agnostic. In other applications, there is often latent structure among tokens, tasks, or embeddings (e.g., syntactic roles, semantic types, or class labels) and MoE have been applied at the embedding, token/sequence or task level. These can be used to define subtask-specific transformations or input conditioning, which suggests that our Omni-Expert approach could be applicable.

## Acknowledgments and Disclosure of Funding

This work was supported by a grant from the National Institutes of Health (R56DC020267-01A1). The NVIDIA Titan V GPU was donated by the NVIDIA Corporation. Ojuba was supported by

the Duke ECE Research Experience for Undergraduates Program. Mainsah received a Microsoft Research Faculty Fellowship.

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

# Appendix

## Table of Contents

# A  Technical Appendices

## A.1  Reverberation Model and Room Impulse Response (RIR) Characteristics

The reverberant signal $x_{\text{rev}}(t)$ is modeled as the convolution of the clean, anechoic speech signal, $s(t)$, with the room impulse response (RIR), $h(t)$:

$$x_{\text{rev}}(t) = s(t) * h(t) \tag{7}$$

To isolate the direct-path component, the RIR is decomposed into two parts:

$$x_{\text{rev}}(t) = s(t) * h_{\text{direct}}(t) + s(t) * h_{\text{reverb}}(t)$$
$$= x_{\text{direct}}(t) + s(t) * h_{\text{reverb}}(t) \tag{8}$$
$$l(t) = x_{\text{rev}}(t) - x_{\text{direct}}(t) \tag{9}$$

where $h_{\text{direct}}(t)$ represents the RIR function for direct path (and early reflections); $h_{\text{reverb}}(t)$ is the RIR of the remaining late reverberation; $x_{\text{direct}}(t)$ represents the direct path signal; and $l(t)$ represents the late reverberant reflections, i.e., the difference between the reverberant and direct path signals.

Table A1 lists characteristics of recorded RIRs of four rooms in the Aachen Impulse Response database [49] used for testing. RIRs were selected from an office, a lecture, a stairway, and a church. For the stairway and the church, RIRs were selected at an azimuth of 90 degrees, where the source and receiver are directly facing each other. RIRs were filtered using an anti-aliasing filter and then downsampled from 48 to 16 kHz before convolution with anechoic speech stimuli. Reverberation times ($RT_{60}$s) were calculated using the Schroeder method [68] using the code provided by [69]. The direct-to-reverberant ratios (DRRs) of the recorded RIRs were calculated using [70].

Table A1: Room impulse response characteristics of test room conditions. $RT_{60}$(s), reverberation time; DRR; Direct-to-reverberant ratio.

| Dataset | Room | Dimensions (L x W x H) (m) | Source Receiver Distance (m) | $RT_{60}$(s) | DRR (dB) |
|---------|------|----------------------------|------------------------------|--------------|----------|
| Aachen Impulse Response (AIR) | Office | 5.0 x 6.4 x 2.9 | 3.0 | 0.6 | 0.4 |
| | Lecture | 10.8 x 10.9 x 3.15 | 5.56 | 0.9 | -0.1 |
| | Stairway | 7.0 x 5.2 | 3.0 | 0.9 | 1.6 |
| | Church | 19.0 x 30.0 | 5.0 | 6.5 | -0.6 |

## A.2    Frame-wise Phoneme Labels

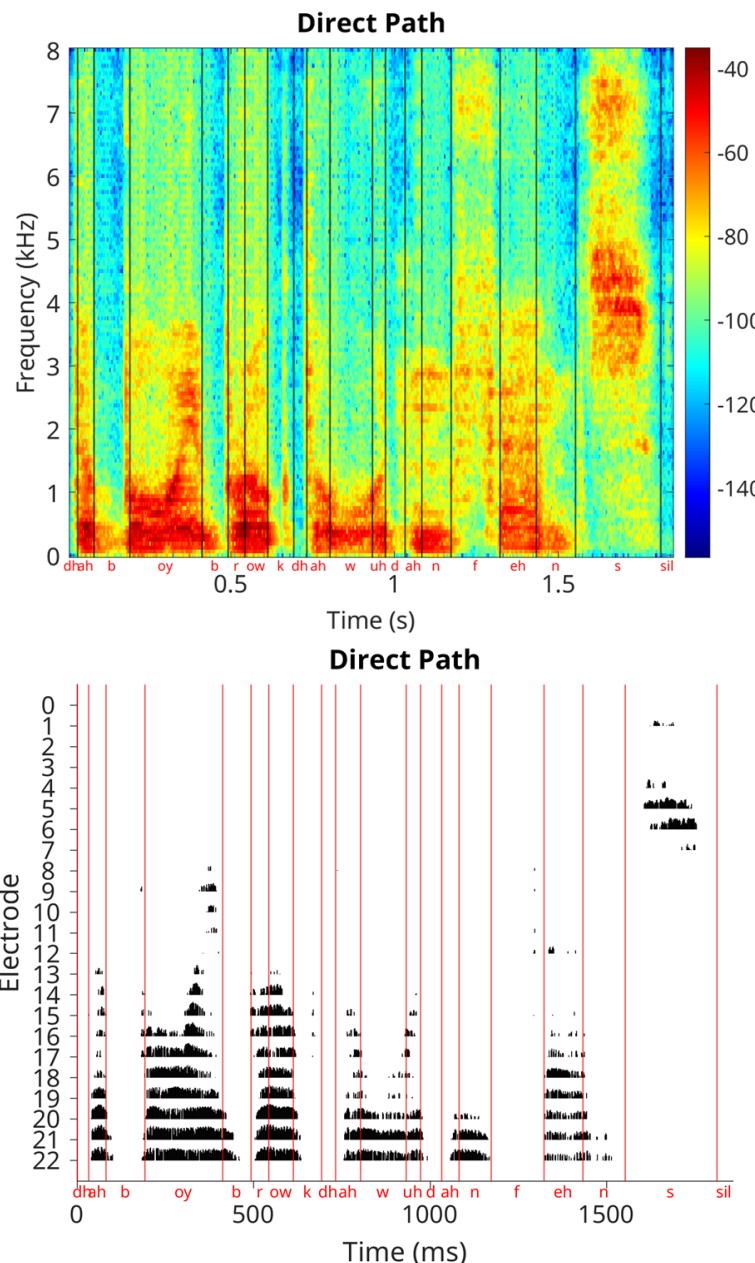

Figure A2.1: Example annotations of phoneme labels aligned to cochlear implant time bins.

### A.3 Example Spectrograms and Electrodograms - LSTM models

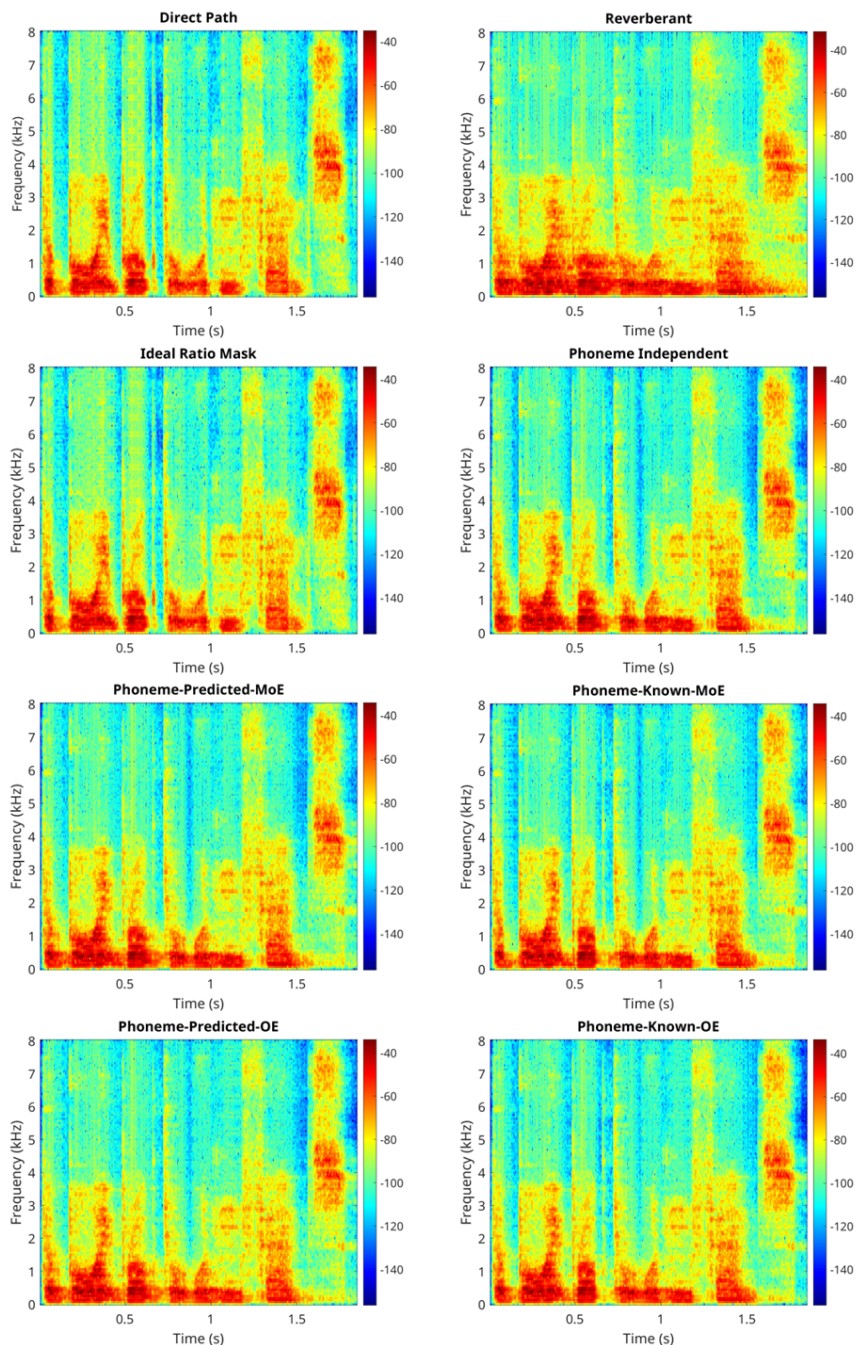

Figure A3.1: Spectrograms of the speech utterance *"the boy broke the wooden fence"* generated for direct path speech, reverberant speech, enhanced reverberant speech after applying the ideal ratio mask and estimated masks with the phoneme independent model, mixture-of-experts (MoE) model with predicted and known phonemes, and Omni-Expert (OE) model with predicted and known phonemes.

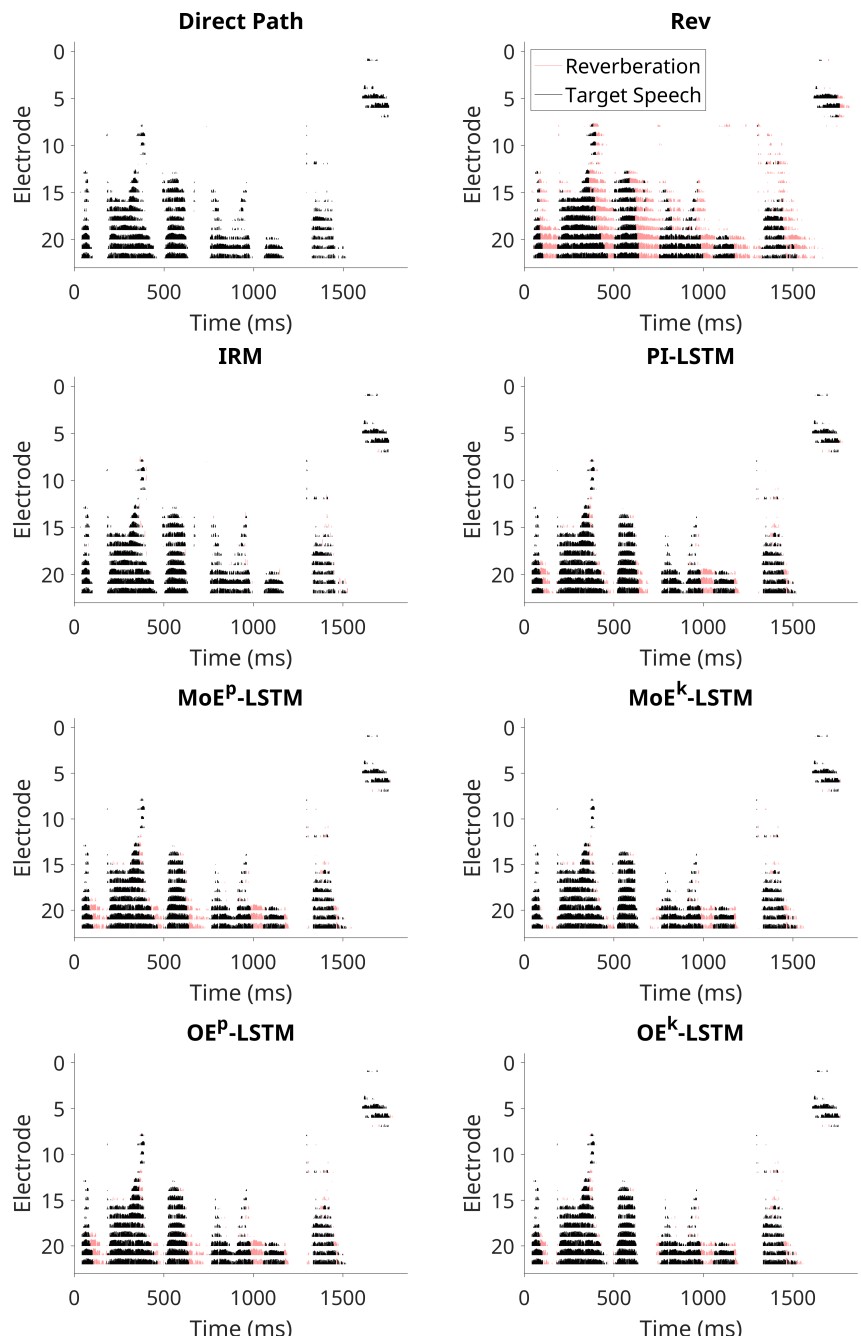

Figure A3.2: Electrodograms of the speech utterance *"the boy broke the wooden fence"* generated for direct path speech, reverberant speech (Rev), enhanced reverberant speech after applying the ideal ratio mask (IRM) and estimated masks with the phoneme independent model (PI), mixture-of-experts model with predicted and known phonemes ($MoE^{p/k}$), and Omni-Expert model with predicted and known phonemes ($OE^{p/k}$).

## A.4 Complexity Analysis

Table A4.1: Summary of complexity of long short-term memory (LSTM) models used for speech dereverberation in cochlear implants.

| Model | Parameters | Training Time[‡] | MACs (M) | Size (MB) |
|---|---|---|---|---|
| Phoneme Independent | 108,225 | 2 hrs 58 mins | 109.44 | 0.43 |
| Phoneme Classifier (PC) | 98440 | 3 hrs | 99.63 | 0.39 |
| Mixture of Experts (MoE) | 40*108,225 + PC | 5 hrs 22 mins | 4377.6 + PC | 16.51 + PC |
| Omni-Expert (OE) | 113555 + PC | 1 hr 57 mins | 109.45 + PC | 0.45 + PC |
| Expert | 108,225 | | | 0.43 |
| Shift + Scale Factors[†] | 2,665 + 2,665 | | | 0.1 + 0.1 |

[†]Shift and scale multilayer perceptrons are not deployed during inference; [‡]NVIDIA Titan V GPU

Table A4.2: Summary of complexity of gated recurrent unit + attention (GRU+A) models used for speech dereverberation in cochlear implants.

| Model | Parameters | Training Time[‡] | MACs (M) | Size (MB) |
|---|---|---|---|---|
| Phoneme Independent (PI) | 127946 | 3 hrs 43 mins | 127.76 | 0.51 |
| Phoneme Classifier (PC) | 124996 | 3 hrs 15 mins | 124.84 | 0.5 |
| Mixture of Experts (MoE) | 40*127946 + PC | 10 hrs 47 mins | 5110.58 + PC | 19.52 + PC |
| Omni-Expert (OE) | 133276 + PC | 1 hr 21 mins | 127.77 + PC | 0.53 + PC |
| -Expert | 127946 | | | 0.51 |
| -Shift + Scale Factors[†] | 2,665 + 2,665 | | | 0.1 + 0.1 |

[†]Shift and scale multilayer perceptrons are not deployed during inference; [‡]NVIDIA Titan V GPU

## A.5 Phoneme Analysis

### A.5.1 Phoneme Distribution

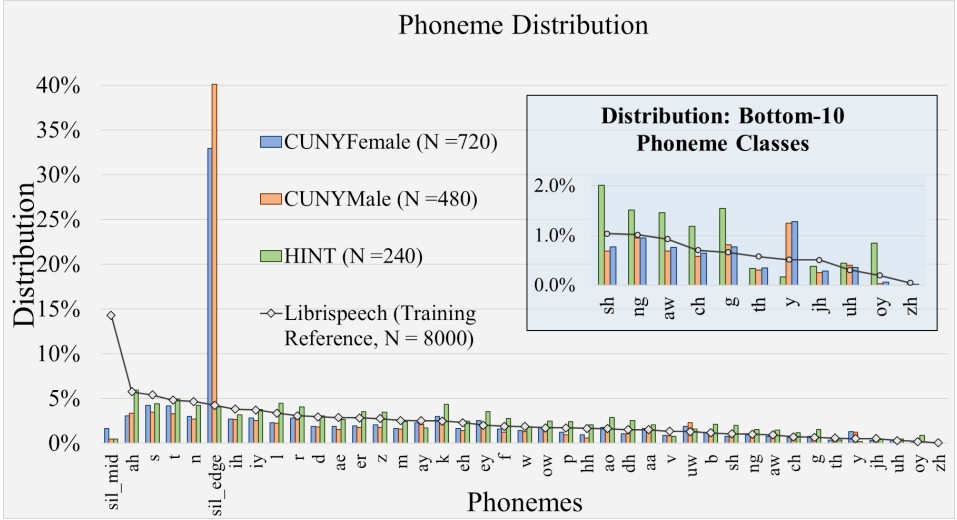

Figure A5.1: Distribution of phoneme classes in the training and test speech datasets sorted by the frequency of phonemes in the training dataset. The silence class (sil) has been divided into two categories: silences occurring at the beginning and end of an utterance, denoted by sil_edge; and silences occurring in the middle of an utterance, denoted by sil_mid. $N$ is the number of speech files in each dataset.

### A.5.2 Phoneme Classifier - LSTM

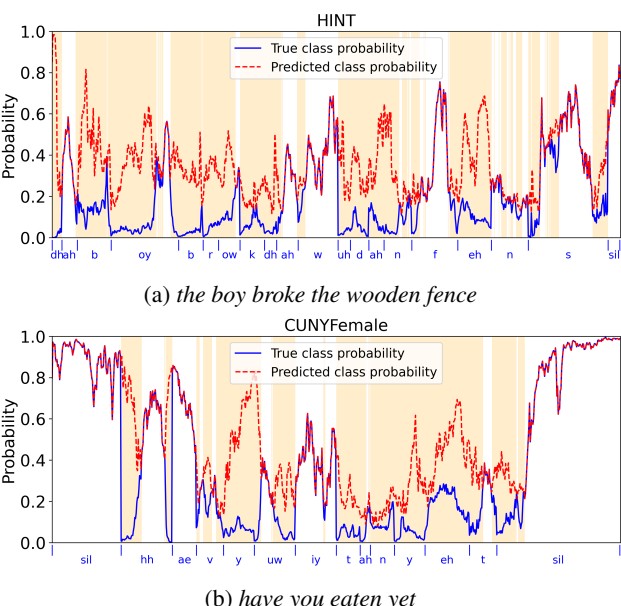

(a) *the boy broke the wooden fence*

(b) *have you eaten yet*

Figure A5.2.1: Framewise probabilities of true and predicted phoneme classes of reverberant speech utterances from (a) HINT and (b) CUNYFemale datasets using the long short-term memory (LSTM) classifier. True class labels are annotated on the $x$-axis. Shaded regions indicate incorrect predictions.

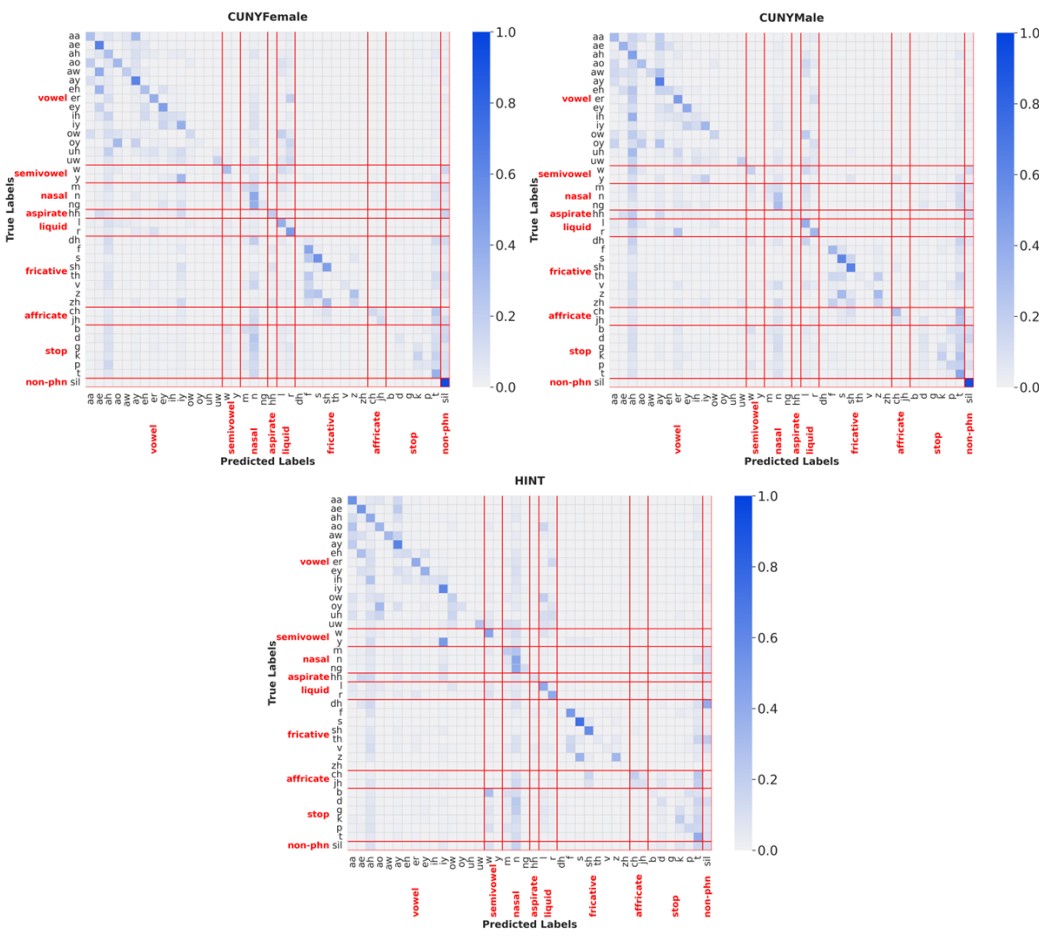

Figure A5.2.2: Confusion matrices of phoneme predictions in test datasets using long short-term memory (LSTM) classifier. Phonemes are annotated by manner of articulation [71].

### A.5.3    Phoneme Classifier - GRU+A

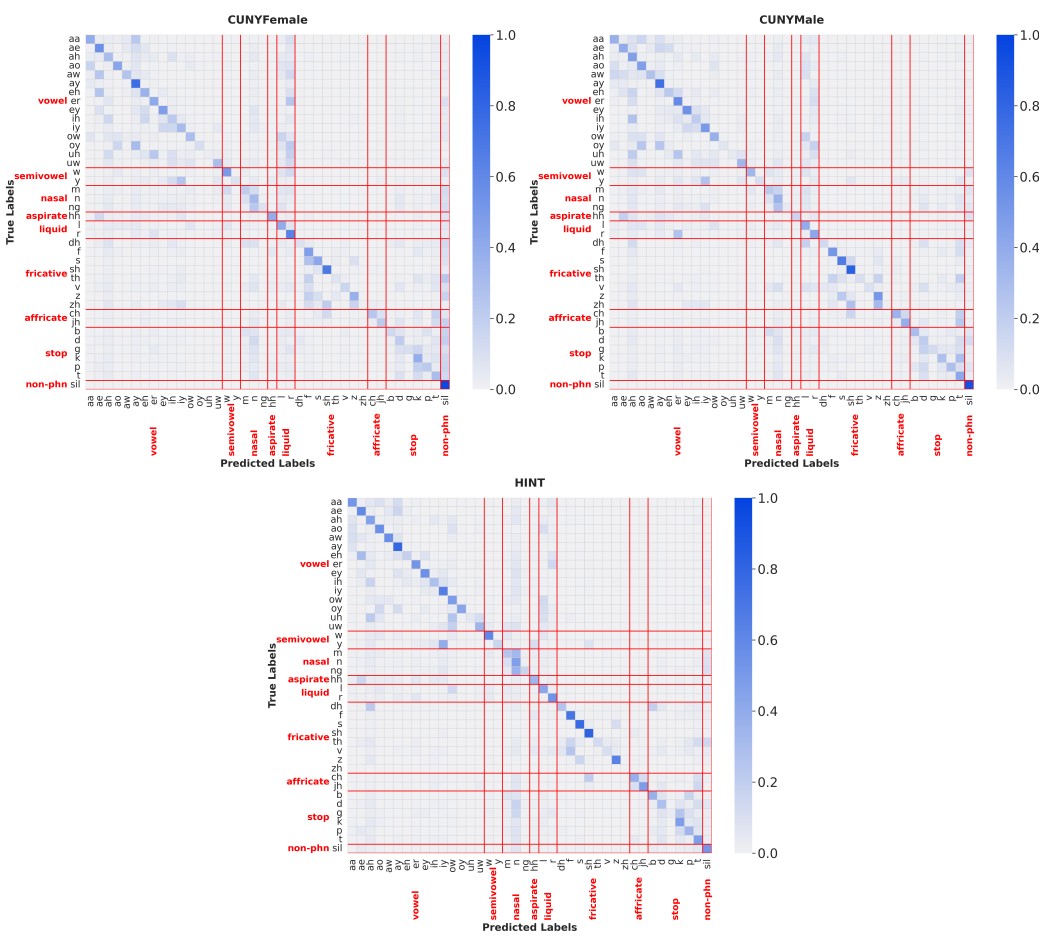

Figure A5.3.1: Confusion matrices of phoneme predictions with GRU+A phoneme classifier in test datasets across room conditions. Phonemes are annotated by manner of articulation [71].

## A.6    Room-specific Mask Estimation Results

### A.6.1    Signal Loss

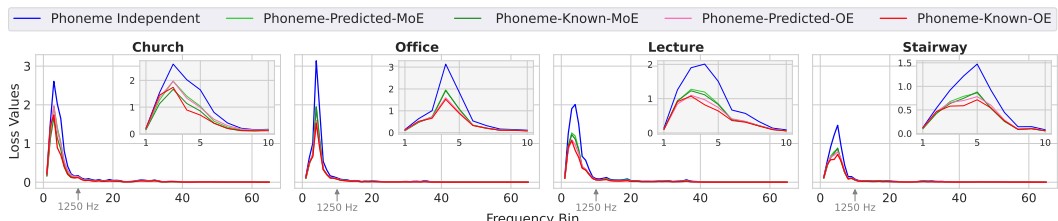

Figure A6.1.1: Average signal loss values across frequency bins with phoneme independent mask estimation and mask estimation with mixture-of-experts (MoE) and Omni-Expert (OE) models with predicted and ideal phoneme knowledge using long short-term memory (LSTM) models.

### A.6.2    Mask Estimation-LSTM models

Table A6.2.1: Long short-term memory (LSTM) models. Mean $\pm$ 95% confidence interval of objective speech intelligibility scores for the reverberant signal (Rev), direct path signal (DP), and across different mask estimation methods: ideal ratio mask (IRM), phoneme independent model (PI), phoneme-based mask predicted by mixture-of-experts/Omni-Expert with ideal phoneme knowledge ($MoE^k$/ $OE^k$), and using predicted phoneme classifier probabilities ($MoE^p$/ $OE^p$). Bold indicates the highest performance among the non-oracle models.

| | SRMR-CI | | | |
| --- | --- | --- | --- | --- |
| | Church | Office | Lecture | Stairway |
| Rev | 0.923 ±0.005 | 1.495 ±0.010 | 1.333 ±0.008 | 1.456 ±0.010 |
| PI | 1.327 ±0.010 | 1.984 ±0.015 | 1.694 ±0.012 | 1.928 ±0.015 |
| $MoE^p$ | 1.351 ±0.010 | 1.956 ±0.014 | 1.731 ±0.012 | 1.939 ±0.015 |
| $MoE^k$ | 1.547 ±0.012 | 1.988 ±0.014 | 1.826 ±0.014 | 2.004 ±0.016 |
| $OE^p$ | **1.370** **±0.010** | **2.029** **±0.015** | **1.787** **±0.013** | **1.990** **±0.016** |
| $OE^k$ | 1.616 ±0.013 | 2.077 ±0.015 | 1.956 ±0.015 | 2.105 ±0.017 |
| IRM | 2.115 ±0.014 | 2.210 ±0.016 | 2.220 ±0.016 | 2.200 ±0.016 |
| DP | 2.438 ±0.017 | 2.423 ±0.018 | 2.452 ±0.018 | 2.473 ±0.017 |

| | STOI | | | |
| --- | --- | --- | --- | --- |
| | Church | Office | Lecture | Stairway |
| Rev | 0.684 ±0.002 | 0.746 ±0.002 | 0.700 ±0.002 | 0.746 ±0.002 |
| PI | 0.771 ±0.002 | 0.814 ±0.002 | 0.779 ±0.002 | 0.823 ±0.002 |
| $MoE^p$ | **0.781** **±0.002** | 0.820 ±0.002 | 0.792 ±0.002 | **0.833** **±0.002** |
| $MoE^k$ | 0.806 ±0.002 | 0.831 ±0.002 | 0.804 ±0.002 | 0.845 ±0.002 |
| $OE^p$ | 0.780 ±0.002 | **0.823** **±0.002** | **0.795** **±0.002** | 0.831 ±0.002 |
| $OE^k$ | 0.819 ±0.002 | 0.845 ±0.002 | 0.827 ±0.002 | 0.855 ±0.001 |
| IRM | 0.975 ±0.000 | 0.970 ±0.001 | 0.974 ±0.000 | 0.969 ±0.001 |
| DP | 1.000 ±0.000 | 1.000 ±0.000 | 1.000 ±0.000 | 1.000 ±0.000 |

### A.6.3   Mask Estimation-GRU+A models

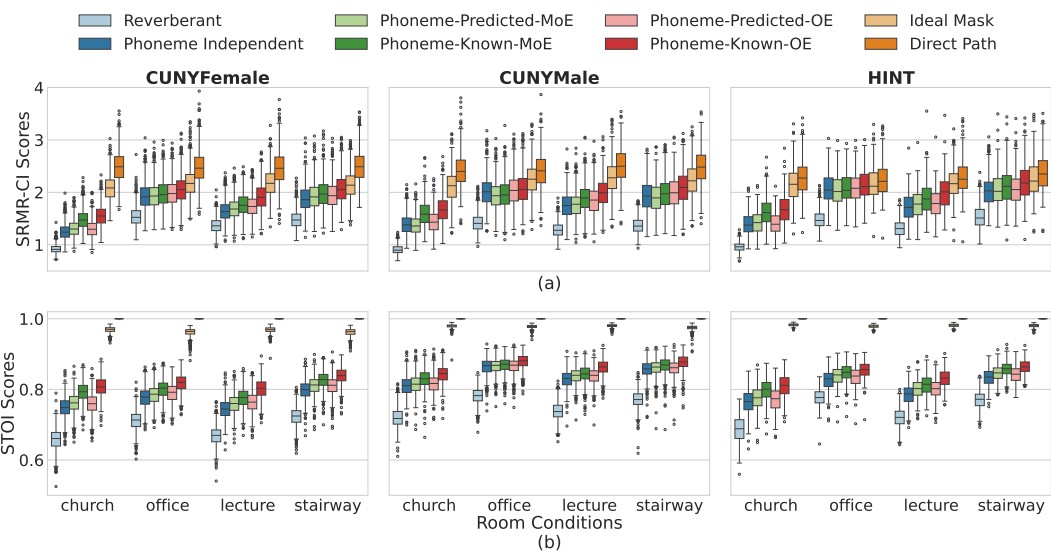

Figure A6.2.1: Objective intelligibility scores of speech from HINT, CUNYFemale, and CUNYMale datasets in church, office, lecture, and stairway rooms. Results are shown for enhanced reverberant speech after applying estimated masks with the phoneme independent model, mixture of experts (MoE) model with predicted and known phonemes, the Omni-Expert (OE) model with predicted and known phonemes, the ideal ratio mask, and the direct path signal. SRMR-CI, Speech-to-reverberation modulation energy ratio for CI users; STOI, short-time objective intelligibility.

## A.7 Additional Ablation Analysis-LSTM models

### A.7.1 Transformation Type

Table A6.3.1: Mean ($\pm$ 95% confidence interval) of objective speech intelligibility scores across different mask estimation methods: phoneme independent model (PI), phoneme-based mask predicted by mixture-of-experts/Omni-Expert with ideal phoneme knowledge ($\text{MoE}^k$/$\text{OE}^k$), and using phoneme classifier probabilities ($\text{MoE}^p$/$\text{OE}^p$). Results are shown for the GRU+Attention (GRU+A) model architecture aggregated across three test datasets in four room conditions. Bold indicates the highest performance among the non-oracle models.

| SRMR-CI | | | | |
|---|---|---|---|---|
| | Church | Office | Lecture | Stairway |
| PI - GRU+A | $1.377 \pm 0.014$ | $2.113 \pm 0.016$ | $1.895 \pm 0.016$ | $2.016 \pm 0.018$ |
| $\text{MoE}^p$ - GRU+A | $1.436 \pm 0.015$ | $2.133 \pm 0.016$ | $1.987 \pm 0.016$ | $2.145 \pm 0.021$ |
| $\text{OE}^p$ - GRU+A | $\mathbf{1.500 \pm 0.013}$ | $\mathbf{2.268 \pm 0.017}$ | $\mathbf{2.059 \pm 0.017}$ | $\mathbf{2.228 \pm 0.019}$ |
| $\text{MoE}^k$ - GRU+A | $1.559 \pm 0.017$ | $2.087 \pm 0.016$ | $1.971 \pm 0.015$ | $2.117 \pm 0.021$ |
| $\text{OE}^k$ - GRU+A | $1.747 \pm 0.014$ | $2.252 \pm 0.017$ | $2.173 \pm 0.017$ | $2.280 \pm 0.019$ |

| STOI | | | | |
|---|---|---|---|---|
| | Church | Office | Lecture | Stairway |
| PI - GRU+A | $0.771 \pm 0.003$ | $0.832 \pm 0.003$ | $0.804 \pm 0.003$ | $0.843 \pm 0.002$ |
| $\text{MoE}^p$ - GRU+A | $0.774 \pm 0.002$ | $0.842 \pm 0.002$ | $0.801 \pm 0.002$ | $0.846 \pm 0.002$ |
| $\text{OE}^p$ - GRU+A | $\mathbf{0.783 \pm 0.002}$ | $\mathbf{0.847 \pm 0.002}$ | $\mathbf{0.825 \pm 0.002}$ | $\mathbf{0.860 \pm 0.002}$ |
| $\text{MoE}^k$ - GRU+A | $0.793 \pm 0.000$ | $0.843 \pm 0.001$ | $0.804 \pm 0.000$ | $0.848 \pm 0.000$ |
| $\text{OE}^k$ - GRU+A | $0.826 \pm 0.002$ | $0.858 \pm 0.002$ | $0.845 \pm 0.002$ | $0.873 \pm 0.001$ |

Table A7.1: Room-specific Objective intelligibility scores (estimated marginal mean ($\pm$ 95% confidence interval)) with the Omni-Expert model with predicted phonemes across different types of feature transformations. Bold indicates the highest performance.

| Speech-to-reverberation modulation energy ratio for CI users (SRMR-CI) | | | | |
|---|---|---|---|---|
| | Church | Office | Lecture | Stairway |
| None | $1.302\ (\pm 0.010)$ | $1.903\ (\pm 0.014)$ | $1.647\ (\pm 0.012)$ | $1.881\ (\pm 0.014)$ |
| Shift Only | $1.278\ (\pm 0.010)$ | $1.968\ (\pm 0.015)$ | $1.695\ (\pm 0.012)$ | $1.903\ (\pm 0.015)$ |
| Scale Only | $1.288\ (\pm 0.009)$ | $1.925\ (\pm 0.014)$ | $1.706\ (\pm 0.012)$ | $1.906\ (\pm 0.015)$ |
| **Scale + Shift** | $\mathbf{1.370\ (\pm 0.010)}$ | $\mathbf{2.029\ (\pm 0.015)}$ | $\mathbf{1.787\ (\pm 0.013)}$ | $\mathbf{1.990\ (\pm 0.016)}$ |

| Short-time objective intelligibility (STOI) | | | | |
|---|---|---|---|---|
| | Church | Office | Lecture | Stairway |
| None | $0.767\ (\pm 0.002)$ | $0.811\ (\pm 0.002)$ | $0.774\ (\pm 0.003)$ | $0.821\ (\pm 0.002)$ |
| Shift Only | $0.767\ (\pm 0.002)$ | $0.809\ (\pm 0.002)$ | $0.774\ (\pm 0.002)$ | $0.815\ (\pm 0.002)$ |
| Scale Only | $0.766\ (\pm 0.002)$ | $0.810\ (\pm 0.002)$ | $0.778\ (\pm 0.002)$ | $0.819\ (\pm 0.002)$ |
| **Scale + Shift** | $\mathbf{0.780\ (\pm 0.002)}$ | $\mathbf{0.823\ (\pm 0.002)}$ | $\mathbf{0.795\ (\pm 0.002)}$ | $\mathbf{0.831\ (\pm 0.002)}$ |

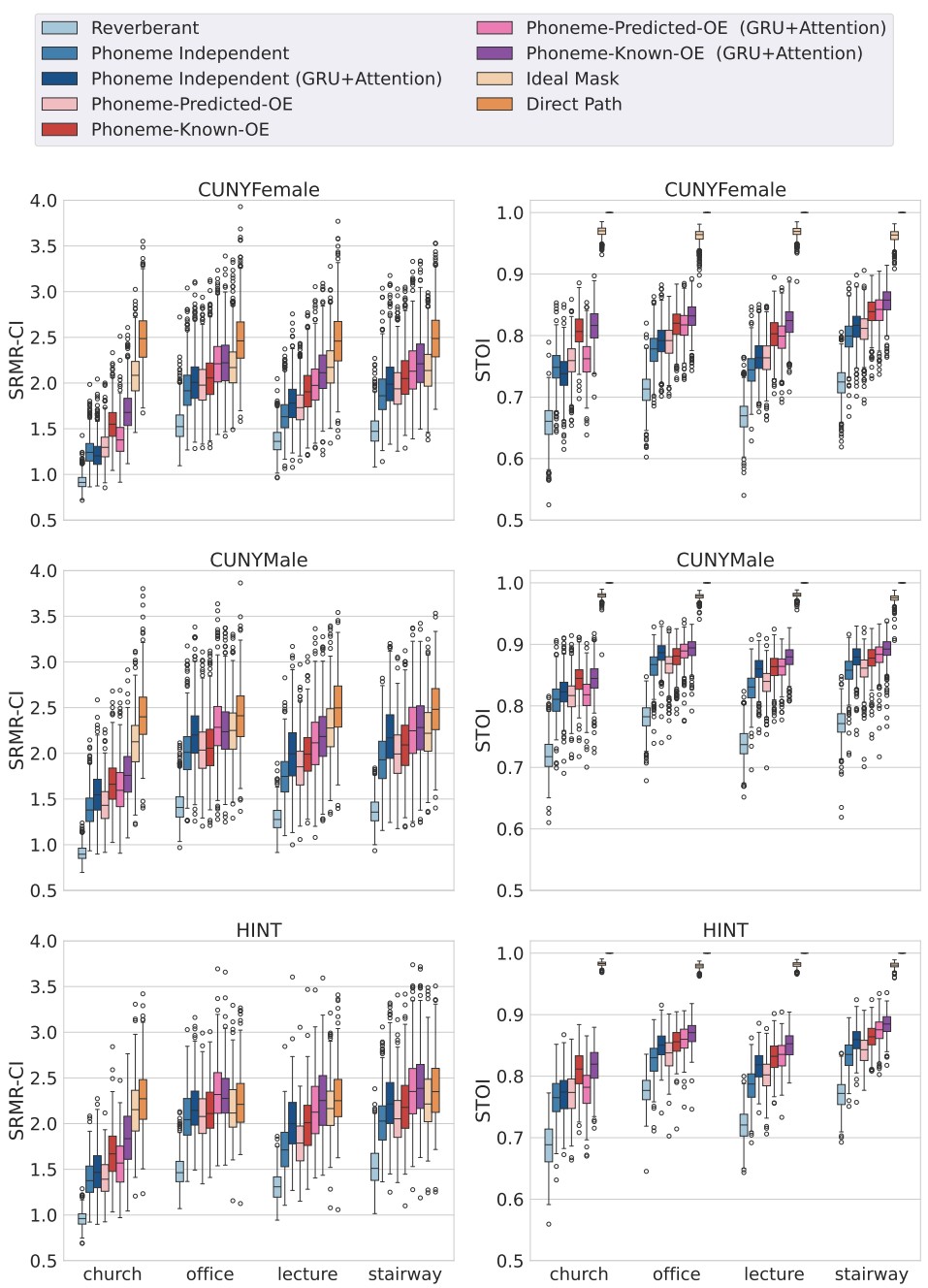

Figure A6.3.1: Boxplots of objective speech intelligibility scores of cochlear implant vocoded speech evaluated for three test datasets in all four room conditions using ratio masks with baseline LSTM and a GRU+Attention networks. Objective speech intellibility measures include speech-to-reverberation modulation energy ratio for CI users (SRMR-CI) and short-time objective intelligibility (STOI). Results are shown for direct path, reverberant speech, enhanced reverberant speech after applying the ideal ratio mask and estimated masks with the Phoneme Independent model and Omni-Expert model (OE) with predicted and known phonemes.

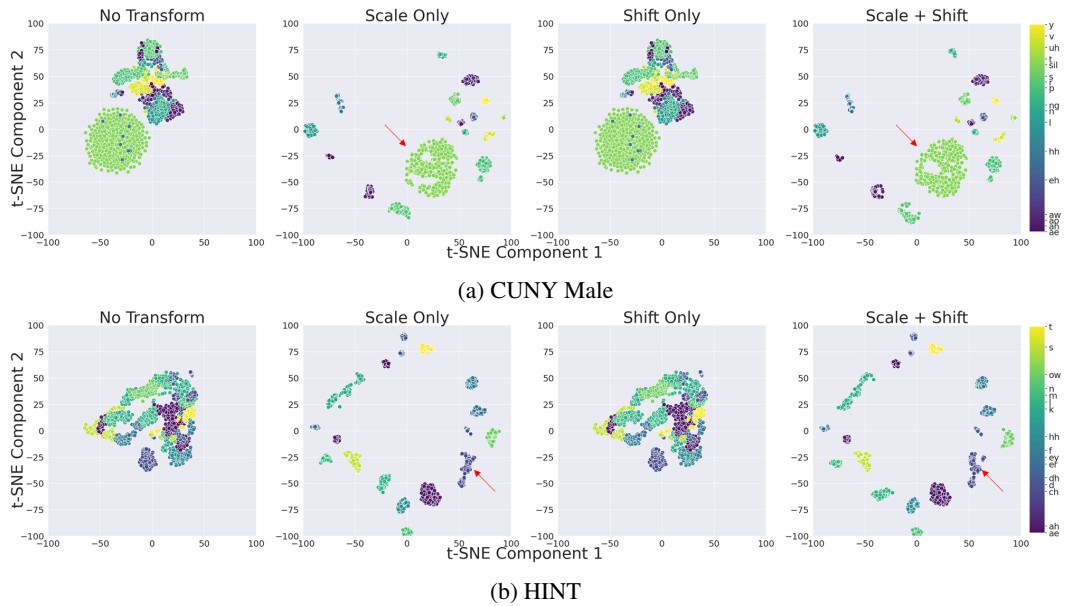

(a) CUNY Male

(b) HINT

Figure A7.1.1: Visualization of phoneme-specific features from a subset of randomly selected phoneme frames (N = 1000) of reverberant speech from the CUNY Male and HINT speech datasets in the stairway room. Column panels represent features: before applying transformations; with scale-only; shift-only; and scale + shift transformations. Arrows indicate an example of visually discernable impact of a shift transformation on a phoneme cluster. t-distributed stochastic neighbor embedding (t-SNE) was used.

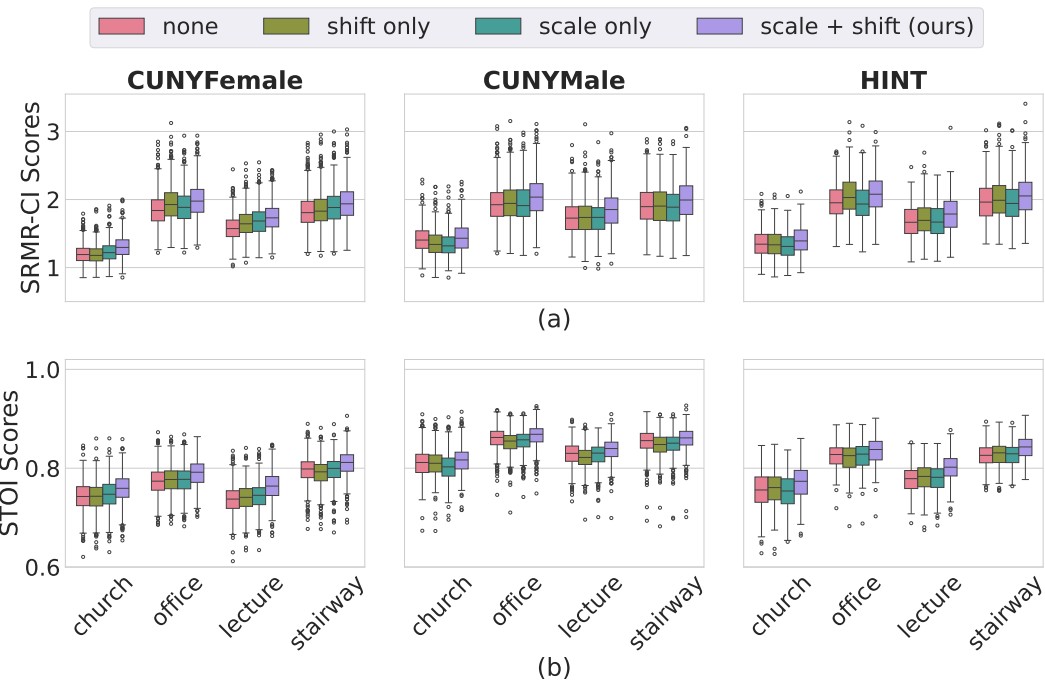

Figure A7.1.2: Boxplots of (a) SRMR-CI and (b) STOI scores evaluated for three test datasets in all four room conditions without any feature modulation and using three different feature modulation techniques: shift only, scale only, and scale+shift (default). Results are shown for the Omni-Expert model with predicted phonemes.

## A.7.2 Position

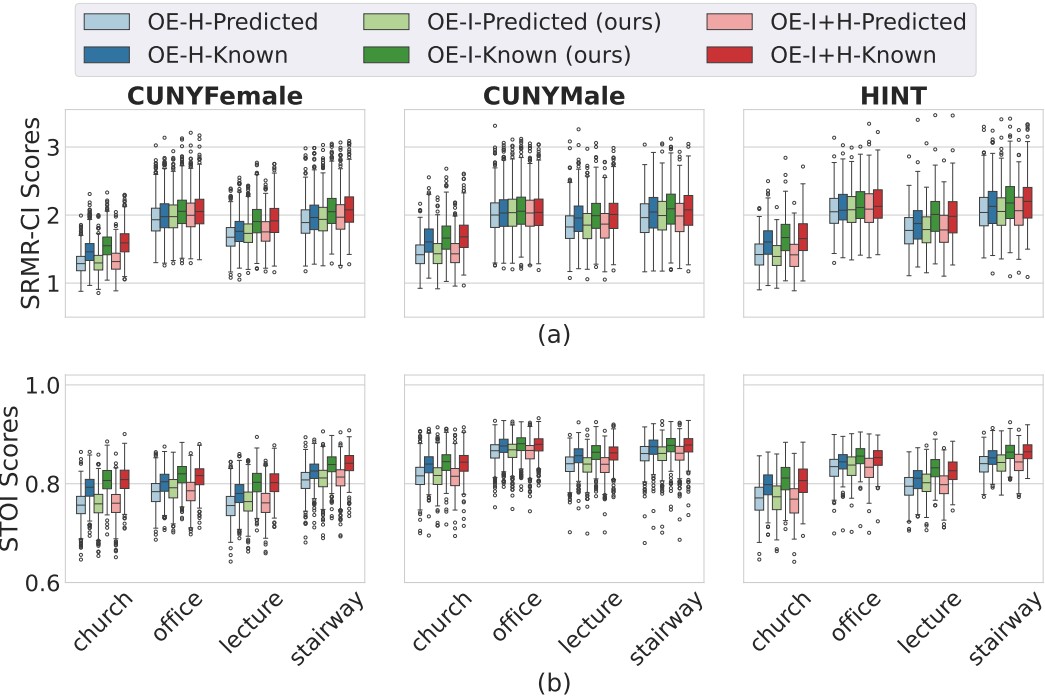

Figure A7.2.1: Boxplots of (a) SRMR-CI and (b) STOI scores evaluated for three test datasets in all four room conditions using ratio masks for an LSTM network of 1 layer. Results are shown for the Omni-Expert model with predicted phonemes.

Table A7.2: Performance across different feature transformation locations, Estimated Marginal Mean (± 95% Confidence interval). Bold indicates the highest performance among the feature transformation locations.

| SRMR-CI | | | | | |
|---|---|---|---|---|---|
| | InsertionPoint | Church | Office | Lecture | Stairway |
| **Phoneme -Predicted -OE** | Hidden Layer (H) | 1.364 (±0.010) | 1.988 (±0.014) | 1.750 (±0.013) | 1.952 (±0.015) |
| | Input Layer (I) | 1.370 (±0.010) | 2.029 (±0.015) | 1.787 (±0.013) | 1.990 (±0.016) |
| | I + H | **1.385 (±0.010)** | **2.039 (±0.015)** | **1.801 (±0.013)** | **1.996 (±0.015)** |
| **Phoneme -Known -OE** | H | 1.544 (±0.013) | 2.030 (±0.015) | 1.849 (±0.014) | 2.028 (±0.016) |
| | I | 1.616 (±0.013) | **2.077 (±0.015)** | 1.956 (±0.015) | 2.105 (±0.017) |
| | I + H | **1.643 (±0.013)** | 2.076 (±0.015) | **1.960 (±0.015)** | **2.109 (±0.017)** |

| STOI | | | | | |
|---|---|---|---|---|---|
| | Insertion Point | Church | Office | Lecture | Stairway |
| **Phoneme -Predicted -OE** | H | 0.778 (±0.002) | 0.818 (±0.002) | 0.789 (±0.002) | 0.829 (±0.002) |
| | I | **0.780 (±0.002)** | **0.823 (±0.002)** | **0.795 (±0.002)** | 0.831 (±0.002) |
| | I + H | 0.778 (±0.002) | 0.818 (±0.002) | 0.793 (±0.002) | **0.833 (±0.002)** |
| **Phoneme -Known -OE** | H | 0.808 (±0.002) | 0.833 (±0.002) | 0.810 (±0.002) | 0.845 (±0.002) |
| | I | **0.819 (±0.002)** | **0.845 (±0.002)** | **0.827 (±0.002)** | 0.855 (±0.001) |
| | I + H | **0.819 (±0.002)** | 0.841 (±0.002) | 0.825 (±0.002) | **0.857 (±0.001)** |

## A.8 Robustness in Noise

**Noisy-Reverberant Testing Conditions**    The test datasets were developed by adding noise from DEMAND [72] and Cocktail Party [73] noise datasets. Two different noise conditions were chosen from DEMAND - Domestic and Public. Domestic noises include kitchen, living room, and washing machine noise environments, and Public noises include the interiors of a cafeteria, restaurant, and a busy subway station. Two-talker Babble (TTB) was selected from Cocktail Party dataset. We used speech from the HINT dataset. Noise was added at signal-to-noise (SNR) levels: -5, 0, 5, 10, 15, 20, and noisy speech was convolved with room impulse responses (RIRs) from office, stairway, lecture, and church room conditions.

### A.8.1   Roomwise model performance - LSTM

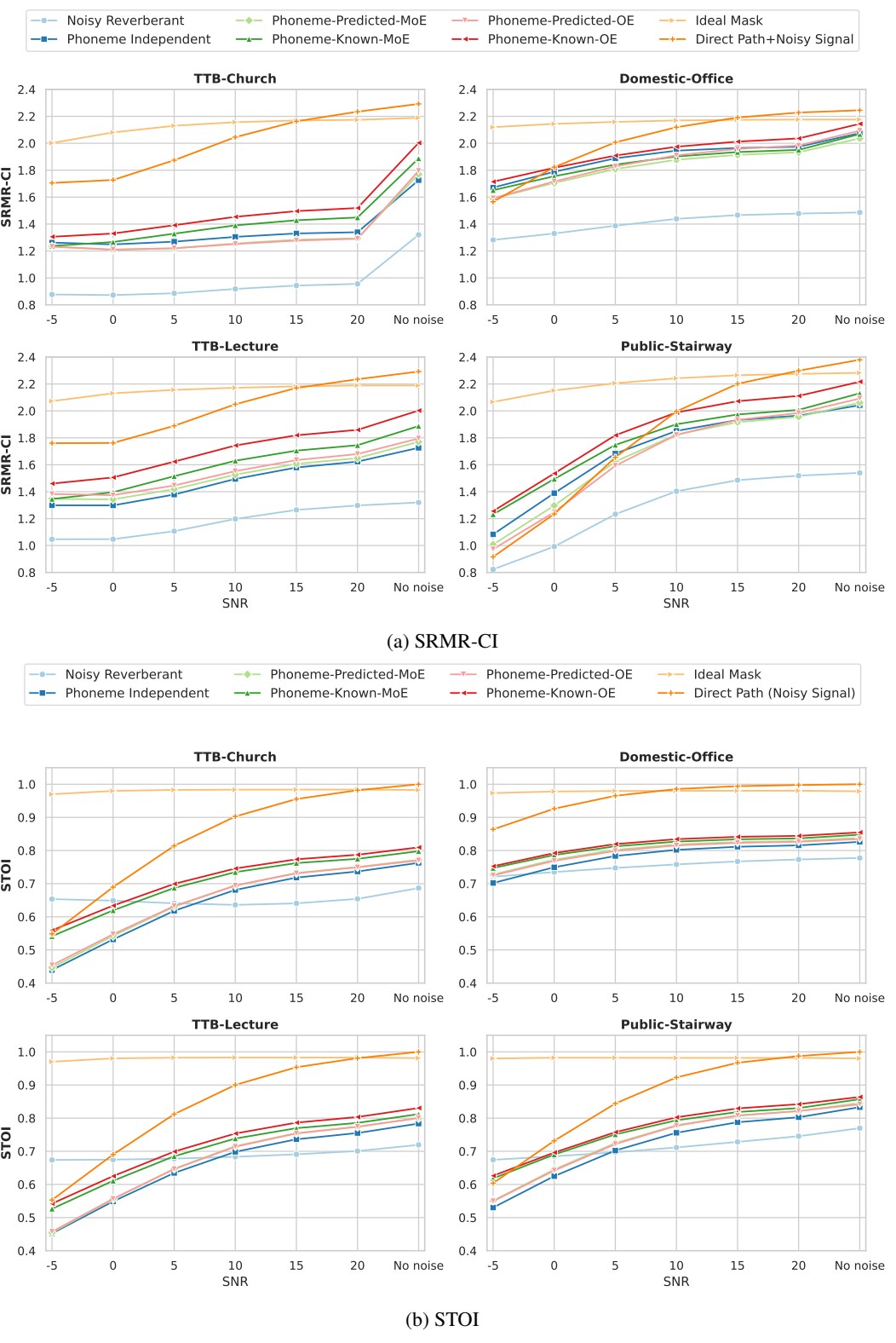

(a) SRMR-CI

(b) STOI

Figure A8.1: SRMR-CI and STOI scores for HINT speech with noise conditions, Domestic and Public noises from DEMAND dataset [72] and Two-Talker Babble (TTB) from Cocktail Party dataset [73] convolved with office, stairway, lecture, and church room conditions (RIRs), respectively. Results are shown for unenhanced noisy reverberant speech, mask estimated using phoneme independent models, phoneme-specific mixture-of-expert model (MoE), phoneme-specific Omni-Expert model (OE), ideal ratio mask (IRM), and the direct path (DP) of the noisy reverberant signal. Noise was added at SNR levels: -5, 0, 5, 10, 15, 20. Additionally, results are shown for no noise (only RIR) condition.

Table A8.1: Performance across different mask estimation methods. Estimated Marginal Mean ($\pm$ 95% Confidence interval) for unenhanced noisy reverberant (Noisy Rev) speech, mask estimated using phoneme independent (PI) model, phoneme-specific mixture-of-expert model ($\text{MoE}^{p/k}$) and phoneme-specific Omni-Expert model ($\text{OE}^{p/k}$) with predicted/known phonemes, ideal ratio mask (IRM), and the direct path of the noisy reverberant signal ($\text{DP}_{noisy}$) across noise conditions (SNR in dB). Bold indicates the highest performance among the non-oracle models.

| | | | | SRMR-CI | | | |
|---|---|---|---|---|---|---|---|
| **Model** | **-5** | **0** | **5** | **10** | **15** | **20** | **No noise** |
| Noisy Rev | 1.007 $\pm$0.015 | 1.060 $\pm$0.014 | 1.153 $\pm$0.015 | 1.239 $\pm$0.017 | 1.290 $\pm$0.018 | 1.313 $\pm$0.018 | 1.327 $\pm$0.018 |
| PI | **1.329** **$\pm$0.022** | **1.431** **$\pm$0.022** | **1.555** **$\pm$0.023** | **1.649** **$\pm$0.024** | **1.702** **$\pm$0.024** | 1.725 $\pm$0.024 | 1.812 $\pm$0.025 |
| $\text{MoE}^p$ | 1.294 $\pm$0.021 | 1.388 $\pm$0.020 | 1.518 $\pm$0.021 | 1.621 $\pm$0.023 | 1.680 $\pm$0.024 | 1.708 $\pm$0.024 | 1.825 $\pm$0.024 |
| $\text{MoE}^k$ | 1.367 $\pm$0.019 | 1.478 $\pm$0.020 | 1.608 $\pm$0.021 | 1.706 $\pm$0.023 | 1.760 $\pm$0.023 | 1.788 $\pm$0.024 | 1.930 $\pm$0.023 |
| $\text{OE}^p$ | 1.295 $\pm$0.021 | 1.387 $\pm$0.020 | 1.522 $\pm$0.021 | 1.634 $\pm$0.023 | 1.701 $\pm$0.024 | **1.734** **$\pm$0.025** | **1.848** **$\pm$0.025** |
| $\text{OE}^k$ | 1.434 $\pm$0.021 | 1.547 $\pm$0.021 | 1.686 $\pm$0.022 | 1.790 $\pm$0.023 | 1.850 $\pm$0.024 | 1.881 $\pm$0.024 | 2.011 $\pm$0.024 |
| IRM | 2.065 $\pm$0.021 | 2.127 $\pm$0.021 | 2.163 $\pm$0.021 | 2.185 $\pm$0.021 | 2.198 $\pm$0.021 | 2.204 $\pm$0.021 | 2.203 $\pm$0.021 |
| $\text{DP}_{noisy}$ | 1.487 $\pm$0.031 | 1.637 $\pm$0.026 | 1.856 $\pm$0.023 | 2.053 $\pm$0.021 | 2.181 $\pm$0.021 | 2.249 $\pm$0.021 | 2.303 $\pm$0.022 |

| | | | | STOI | | | |
|---|---|---|---|---|---|---|---|
| **Model** | **-5** | **0** | **5** | **10** | **15** | **20** | **No noise** |
| Noisy Rev | 0.681 $\pm$0.002 | 0.686 $\pm$0.003 | 0.691 $\pm$0.003 | 0.697 $\pm$0.003 | 0.707 $\pm$0.003 | 0.718 $\pm$0.003 | 0.738 $\pm$0.003 |
| PI | 0.531 $\pm$0.009 | 0.614 $\pm$0.008 | 0.685 $\pm$0.006 | 0.734 $\pm$0.005 | 0.763 $\pm$0.004 | 0.778 $\pm$0.003 | 0.801 $\pm$0.003 |
| $\text{MoE}^p$ | 0.545 $\pm$0.009 | **0.629** **$\pm$0.008** | **0.701** **$\pm$0.006** | **0.752** **$\pm$0.005** | **0.780** **$\pm$0.003** | **0.794** **$\pm$0.003** | **0.814** **$\pm$0.003** |
| $\text{MoE}^k$ | 0.608 $\pm$0.007 | 0.677 $\pm$0.006 | 0.734 $\pm$0.005 | 0.773 $\pm$0.004 | 0.796 $\pm$0.003 | 0.807 $\pm$0.003 | 0.829 $\pm$0.002 |
| $\text{OE}^p$ | **0.546** **$\pm$0.009** | **0.629** **$\pm$0.008** | 0.700 $\pm$0.006 | 0.750 $\pm$0.004 | 0.779 $\pm$0.003 | 0.793 $\pm$0.003 | 0.812 $\pm$0.003 |
| $\text{OE}^k$ | 0.620 $\pm$0.007 | 0.687 $\pm$0.006 | 0.744 $\pm$0.005 | 0.784 $\pm$0.004 | 0.808 $\pm$0.003 | 0.819 $\pm$0.002 | 0.840 $\pm$0.002 |
| IRM | 0.973 $\pm$0.001 | 0.980 $\pm$0.001 | 0.982 $\pm$0.000 | 0.982 $\pm$0.000 | 0.982 $\pm$0.000 | 0.982 $\pm$0.000 | 0.981 $\pm$0.000 |
| $\text{DP}_{noisy}$ | 0.642 $\pm$0.010 | 0.759 $\pm$0.008 | 0.859 $\pm$0.005 | 0.928 $\pm$0.003 | 0.968 $\pm$0.002 | 0.987 $\pm$0.001 | 1.000 $\pm$0.000 |

### A.8.2 Roomwise model performance - GRU+A

Table A8.2: Objective speech intelligibility scores (estimated marginal mean ($\pm$ 95% confidence interval) for mask estimated using phoneme independent (PI) model, and phoneme-specific Omni-Expert model with predicted/known phonemes ($OE^{p/k}$) across varying noise types and signal-to-noise ratio (SNR in dB). Results are aggregated for HINT speech with domestic noise + office, public noise + stairway, two-talker babble (TTB) noise + lecture and TTB + church. Results are shown for the base LSTM model and the GRU+Attention (GRU+A) model. Bold indicates the highest performance among the non-oracle models.

| | | | SRMR-CI | | | | |
|---|---|---|---|---|---|---|---|
| **Model** | **-5** | **0** | **5** | **10** | **15** | **20** | **No noise** |
| PI - LSTM | 1.329 ±0.022 | 1.431 ±0.022 | 1.555 ±0.023 | 1.649 ±0.024 | 1.702 ±0.024 | 1.725 ±0.024 | 1.812 ±0.025 |
| PI - GRU+A | 1.353 ±0.024 | 1.486 ± 0.022 | 1.656 ±0.023 | 1.784 ±0.025 | 1.849 ±0.026 | 1.878 ±0.026 | 1.986 ±0.028 |
| $OE^p$ - LSTM | 1.295 ±0.021 | 1.387 ±0.020 | 1.522 ±0.021 | 1.634 ±0.023 | 1.701 ±0.024 | 1.734 ±0.025 | 1.848 ±0.025 |
| $OE^p$ - GRU+A | **1.368 ±0.026** | **1.499 ±0.024** | **1.685 ±0.026** | **1.826 ±0.027** | **1.902 ±0.028** | **1.936 ±0.029** | **2.119 ±0.030** |
| $OE^k$ - LSTM | 1.434 ±0.021 | 1.547 ±0.021 | 1.686 ±0.022 | 1.790 ±0.023 | 1.850 ±0.024 | 1.881 ±0.024 | 2.011 ±0.024 |
| $OE^k$ - GRU+A | 1.555 ±0.023 | 1.706 ±0.023 | 1.864 ±0.025 | 1.972 ±0.026 | 2.027 ±0.027 | 2.052 ±0.027 | 2.217 ±0.027 |

| | | | STOI | | | | |
|---|---|---|---|---|---|---|---|
| **Model** | **-5** | **0** | **5** | **10** | **15** | **20** | **No noise** |
| PI - LSTM | 0.531 ±0.009 | 0.614 ±0.008 | 0.685 ±0.006 | 0.734 ±0.005 | 0.763 ±0.004 | 0.778 ±0.003 | 0.801 ±0.003 |
| PI- GRU+A | 0.558 ±0.009 | 0.642 ±0.007 | 0.716 ±0.006 | 0.767 ±0.004 | 0.795 ±0.003 | 0.808 ±0.003 | 0.823 ±0.003 |
| $OE^p$ - LSTM | 0.546 ±0.009 | 0.629 ±0.008 | 0.700 ±0.006 | 0.750 ±0.004 | 0.779 ±0.003 | 0.793 ±0.003 | 0.812 ±0.003 |
| $OE^p$ - GRU+A | **0.559 ±0.009** | **0.645 ±0.008** | **0.720 ±0.006** | **0.773 ±0.005** | **0.804 ±0.004** | **0.818 ±0.003** | **0.836 ±0.003** |
| $OE^k$ - LSTM | 0.620 ±0.007 | 0.687 ±0.006 | 0.744 ±0.005 | 0.784 ±0.004 | 0.808 ±0.003 | 0.819 ±0.002 | 0.840 ±0.002 |
| $OE^k$ - GRU+A | 0.629 ±0.007 | 0.701 ±0.006 | 0.760 ±0.005 | 0.801 ±0.004 | 0.824 ±0.003 | 0.835 ±0.002 | 0.855 ±0.002 |

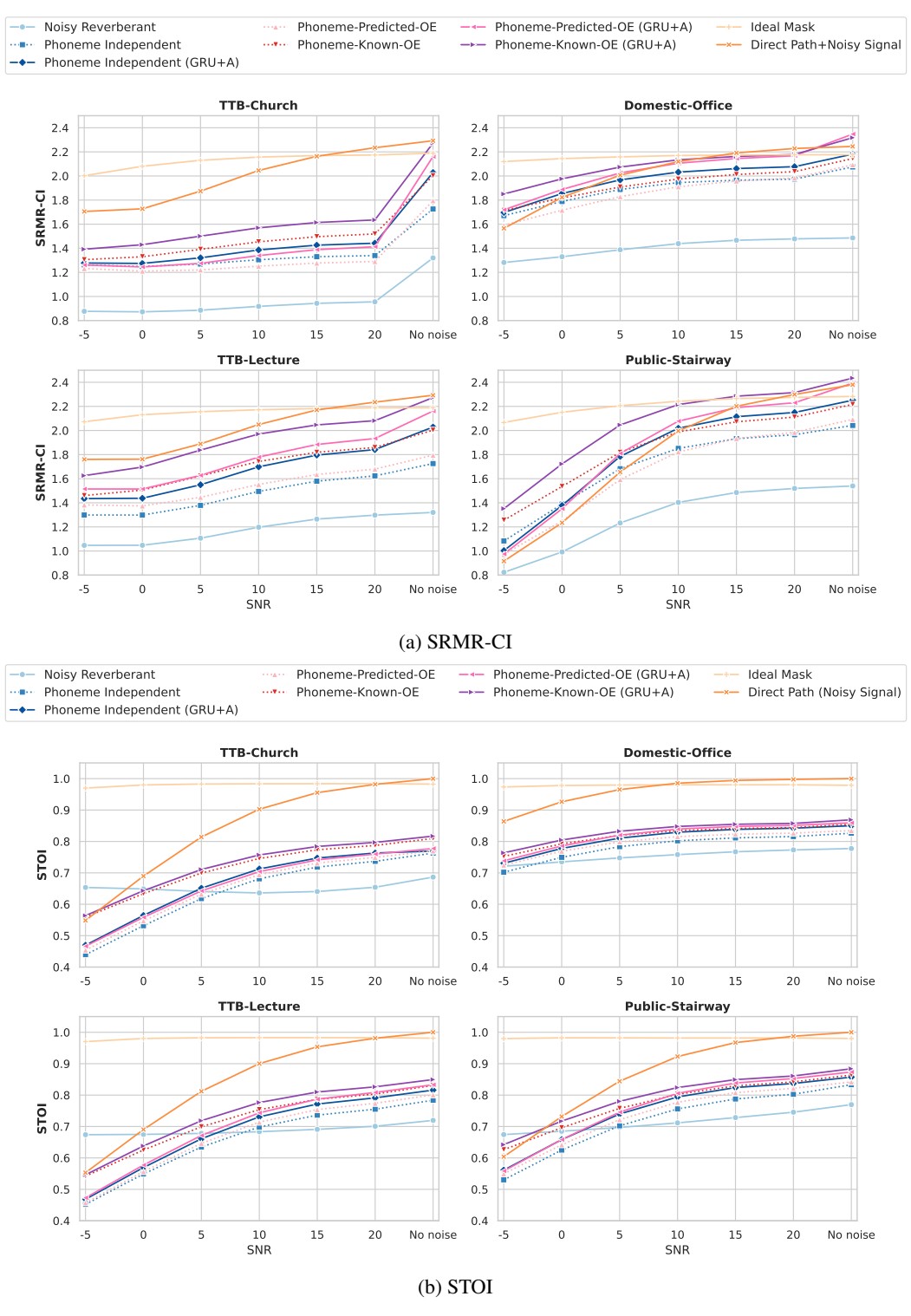

(a) SRMR-CI

(b) STOI

Figure A8.2: SRMR-CI and STOI scores for HINT speech with noise conditions, Domestic and Public noises from DEMAND dataset and Two-Talker Babble (TTB) from Cocktail Party dataset convolved with office, stairway, lecture, and church room conditions (RIRs), respectively. Results are shown for unenhanced noisy reverberant speech, mask estimated using phoneme independent models, phoneme-specific Omni-Expert models (OE) - LSTM and GRU+A, ideal ratio mask (IRM), and the direct path (DP) of the noisy reverberant signal. Noise was added at signal-to-noise (SNR) levels: -5, 0, 5, 10, 15, 20. Additionally, results are shown for no noise (only RIR) condition.

### A.8.3  Room-specific Phoneme Classifier Performance in Noisy Reverberant Conditions

Table A8.3.1: Phoneme classification accuracies in noisy reverberant test conditions using long short-term memory (LSTM) model architecture (%). Models are trained in reverberant only conditions.

| Dataset | -5 | 0 | 5 | 10 | 15 | 20 |
|---|---|---|---|---|---|---|
| HINT-Domestic-Office | 16.4 | 19.87 | 23.43 | 26.04 | 28.18 | 29.76 |
| HINT-Public-Stairway | 6.08 | 9.31 | 14.32 | 19.71 | 24.25 | 27.31 |
| HINT-TTB-Lecture | 7.59 | 10.56 | 14.15 | 18.28 | 22.06 | 24.81 |
| HINT-TTB-Church | 7.57 | 9.93 | 12.70 | 15.78 | 18.43 | 20.23 |

Table A8.3.2: Phoneme Classification Accuracies in noisy reverberant test conditions using gated recurrent unit + attention (GRU+A) model architecture (%). Models are trained in reverberant only conditions.

| Dataset | -5 | 0 | 5 | 10 | 15 | 20 |
|---|---|---|---|---|---|---|
| HINT-Domestic-Office | 24.04 | 29.46 | 34.97 | 40.03 | 43.26 | 45.14 |
| HINT-Public-Stairway | 6.50 | 11.65 | 19.75 | 28.16 | 35.36 | 40.25 |
| HINT-TTB-Lecture | 7.91 | 12.13 | 18.20 | 25.15 | 32.18 | 37.34 |
| HINT-TTB-Church | 7.96 | 11.50 | 16.36 | 21.73 | 26.67 | 29.89 |

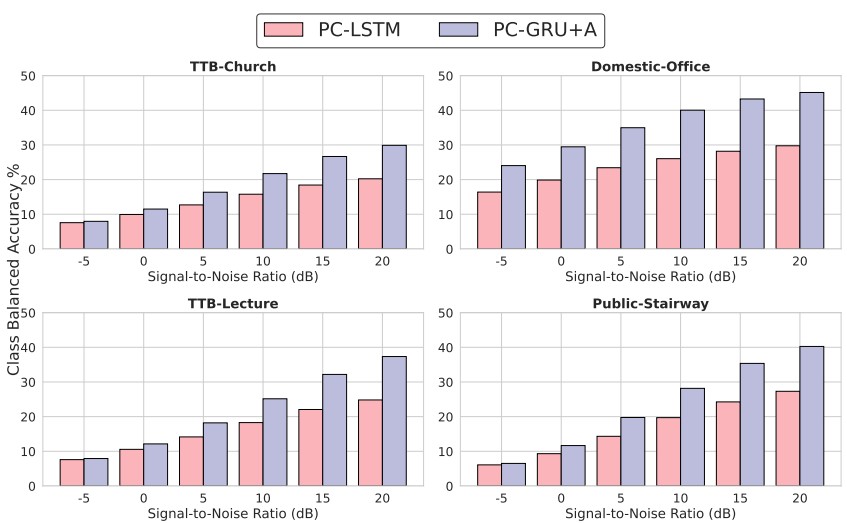

Figure A8.3.1: Phoneme classifier (PC) performance in noisy reverberant room conditions using the long short-term memory (LSTM) model and the gated recurrent unit + attention (GRU+A) architecture. TTB, two-talker babble.

