# OpenReview forum: "The Omni-Expert: A Computationally Efficient Approach to Achieve a Mixture of Experts in a Single Expert Model"
_NeurIPS.cc/2025/Conference — NeurIPS 2025 poster_

### Official Review · Reviewer_5JAh · 2025-06-03

**Clarity:** 3
**Significance:** 4
**Originality:** 3
**Rating:** 5
**Confidence:** 3

**Summary:**

This submission provided a new omni-expert (OE) model, a more efficient alternative to the mixture-of-experts model (MoE) by creating a subtask-specific feature transformation block so that all the features are fed to a single model. The OE model is evaluated against MOE model in the context of cochlear implant (CI), where multiple tasks are performed. The OE model is found to have a better performance and is more computationally effective.

**Questions:**

What are the reasons that the MOE and OE models perform similarly in terms of the STOI metric?

Are the relatively small improvements from MOE and OE models meaningful from using a PI model?

How would other strategies of improving MOE perform in this task, such as merging the expert models (i.e. from existing work "Merging Experts into One: Improving Computational Efficiency of Mixture of Experts")?

**Ethical Concerns:**

["NO or VERY MINOR ethics concerns only"]

**Final Justification:**

The authors' rebuttal sufficiently addressed my previous concerns. The authors clarified that the main selling point of the OE model is reduced computational cost and justified why the STOI metric might fail at capturing improvements. The authors also made the effort to compare with prior work mentioned in my original review. As such, I maintain my original recommendation of acceptance.

While this work is primarily focused on signal processing for cochlear implants, the OE methodology could be adapted to similar types of signals, albeit it would be an entirely new study. I lean towards publishing for the new methodology, even with a potentially limited application case.

**Limitations:**

yes

**Quality:**

2

**Strengths And Weaknesses:**

Quality: The submission presents an OE model architecture, and its performance is rigorously compared to the MOE model variant in the context of phoneme prediction of CI. However, in some of the results, i.e. table 2, where the OE did not show any performance improvement in terms of the STOI metric, or in figure 4, where both the MOE and OE models show minimal improvements to the phoneme independent (PI) model. The submission could benefit from the authors justifying that the minimal improvement is meaningful and discussing the result more, where STOI is not improved by using OE compared to MOE.

Clarity: The article is well-written and organized, although some of the specific terms could use a bit more explanation to be clearer for people outside the field of audio processing field, such as the difference between a phenome-known model and a phenome-predicted model.

Significance: The submission is proposing a new model for CI data processing that is less computationally expensive and performs better. It can be very beneficial for CI manufacturing.

Originality: This submission proposes a new OE model that attempts to improve upon MOE by using a feature transformation block. However, existing work exists that attempts to reduce the computational cost of MOE systems, albeit using a different strategy. (See the paper titled "Merging Experts into One: Improving Computational Efficiency of Mixture of Experts") The authors should mention previous work and consider explaining why this new strategy could be a better approach.

---

> ### Author Rebuttal · Authors · 2025-07-30
>
> Thank you for the feedback and spending your time reviewing our paper. Please find below responses to your comments and the changes we will make to the paper.
> ## C1: Relative Performance Improvements with Objective Speech Intelligibility Metrics.
> > **In some of the results, i.e. table 2, where the OE did not show any performance improvement in terms of the STOI metric, or in figure 4, where both the MOE and OE models show minimal improvements to the phoneme independent (PI) model. The submission could benefit from the authors justifying that the minimal improvement is meaningful and discussing the result more, where STOI is not improved by using OE compared to MOE.**
>
> > **What are the reasons that the MOE and OE models perform similarly in terms of the STOI metric? Are the relatively small improvements from MOE and OE models meaningful from using a PI model?**
>
> - ***STOI excludes frames with silent gaps in clean speech, whereas reverberation occurs in silent speech gaps***. The STOI metric [1] was developed for evaluating intelligibility of speech *in noise*, not reverberation. From the metric authors, the STOI computation does not include silent gaps in the speech signal and focuses only on speech segments. From page 2126, Section II. [1]:
>
>       Before evaluation, silent regions which do not contribute to speech intelligibility are removed. This is done by first finding the frame with the maximum energy of the clean speech signal. Both signals are then reconstructed, excluding all the frames where the clean speech energy is lower than 40 dB with respect to the maximum clean speech energy frame.
>
>   The rationale the authors provide is a metric to reflect intelligibility of actual speech content rather than being influenced by non-informative silent regions. However, reverberant reflections persist during silent gaps. Removal of reverberant speech reflections typically occurs from the tail of decay inward (see electrodograms in ***Figure 4(a)***). Thus, removal of reverberant reflections in the same silent speech gap will not be captured by the STOI metric.
>
> - ***Speech-to-Reverberation Modulation Energy Ratio (SRMR) metric [2,3] was developed specifically for evaluating speech intelligibility in reverberation.*** The original SRMR metric was proposed for normal hearing listeners [2] and later adapted for CI users (SRMR-CI) by using the CI filterbanks [3]. Thus, the SRMR-CI metric provides a more reliable speech intelligibility predictor for cochlear implants in reverberation. Nonetheless, we included STOI for completeness and to facilitate comparison with prior literature. We will include the clarification of the STOI and SRMR metrics in the Performance Evaluation portion (***Section 4.3***) in the revised paper.
>
> - ***Relative performance improvements from PI to OE to MoE.*** We implemented the MoE from [4] who showed improvements in SRMR-CI and STOI scores, as well as intelligibility of CI vocoded speech in normal hearing listeners with the MoE model relative to a PI model (see ***194-196*** in the main paper). As we stated in the paper, our goal with the OE is to achieve at a minimum the same performance at a reduced computational cost relative to a counterpart MoE model (***lines 196-198***, main paper). The OE model offers a computationally efficient alternative to the MOE model. The OE has a much faster training time relative to the MoE model (***Table 5***, main paper). We observe statistically significant improvements in SRMR-CI scores in aggregate (***Table 2***, SRMR-CI) and room-specific (***Figure 4(b)***, SRMR-CI) results from PI to MoE to OE. We have discussed the limitations of the STOI metric not capturing the impact of removal of reverberant reflections in the same silent speech gaps, which likely explains relatively small differences in STOI between PI vs. {MoE, OE} and similar STOI between MoE and OE models (STOI results in ***Table 2 and Figure 4(b)***). Note that we also included results with an advanced model architecture for the PI and OE models in the supplement file (***Section A.9***), where we obtain similar performance trends with wider margins of improvement from PI to OE models (***Figures A9.2.3, A9.2.4 and A9.3.2***). We will consolidate results from both models in the revised paper to show that our OE approach is applicable to other models.
>
> - OE provides a higher performance upperbound vs. MoE. We included performance with perfect sub-task selection with the gating network, i.e., perfect phoneme classification, in the MoE and OE configurations; this provides a benchmark of additional improvements that could be obtained by just improving the phoneme classifier performance with the same mask estimation model. We have additional results in the supplement file that shows increased phoneme classification rates with an advanced phoneme classifier (GRU+Attention) vs. the LSTM phoneme classifier used in the main paper.
>
> ## C2: Definition of Terms.
>
> > **Some of the specific terms could use a bit more explanation to be clearer for people outside the field of audio processing field, such as the difference between a phenome-known model and a phenome-predicted model.**
>
> We will clarify the definitions of “phoneme-known” and “phoneme-predicted” models in the *Performance Evaluation* section of the paper (***section 4.3, lines 198-199***). The phoneme-known model refers to a MoE/OE mask estimation with perfect phoneme classification (i.e., oracle condition with ideal expert/subtask selection). The phoneme-predicted model refers to the MoE/OE mask estimation with phoneme classifier predictions that are used to probabilistic weight the expert/subtask-specific outputs during inference.
>
> ## C3: Relationship to Prior Work
>
> > **Existing work exists that attempts to reduce the computational cost of MOE systems, albeit using a different strategy. (See the paper titled "Merging Experts into One: Improving Computational Efficiency of Mixture of Experts") The authors should mention previous work and consider explaining why this new strategy could be a better approach.**
>
> > **How would other strategies of improving MOE perform in this task, such as merging the expert models (i.e. from existing work "Merging Experts into One: Improving Computational Efficiency of Mixture of Experts")?**
>
> We thank the reviewer for highlighting the related work, “Merging Experts into One: Improving Computational Efficiency of Mixture of Experts” [5], which we will cite and discuss in the revised manuscript in the Related Work section. Merging of Experts into One (MEO) addresses computational efficiency in MoE by aggregating multiple expert parameters at inference time using gating weights accordingly:
>
> $\hat{W}_i = \sum\_ {k \in T} G_k(x_i ) \cdot W_k, \hat{\beta} _i  = \sum\_ {k \in T} G_k(x_i ) \cdot \beta_k$   (1)
>
> $y_i = \sigma(\hat{W}_ix_i + \hat{\beta} _i)$  (2)
>
> where $x_i$ is the input; $G_k(x_i)$, is the gating network probability for input $x_i$ of the $k$-th expert; $W_k$ and $\beta_k$ represent the weight and bias terms of the $k$-th expert; $\hat{W}_i $ ,$\hat{\beta}_i$ are the aggregate weight and bias terms, respectively, for $x_i$; $T$ is the total number of selected experts. The final output for MEO is given by $y_i$, where $\sigma$ represents the activation function.
>
> While the MEO reduces the computation cost at inference when compared to evaluating all selected experts, it still requires training and storing ***multiple expert*** networks, as well as performing parameter aggregation dynamically during inference. In contrast, our proposed Omni-Expert (OE) model utilizes ***a single expert*** network, $E$, throughout both training and inference accordingly:
>
> $z_k = a_k⊙x_i+b_k$ (3)
>
> $y_i =\sum\_ {k \in T} G_k(x_i )\cdot E(z_k)$ (4)
>
> where $x_i$ is the input feature vector; $z_k$ is the transformed feature vector; $a_k$ and $b_k$ are the scale and shifting factor vectors, respectively, for the $k$-th subtask; and ⊙ is element-wise multiplication.  Instead of selecting or merging experts, we apply subtask-specific (e.g., in this case, phoneme-based) transformations to the input features (3), followed by a gating network that softly weights the output of each transformed input (4). This allows our model to retain the representational flexibility of MoE systems while avoiding the cost of loading or merging multiple experts at inference, making it significantly more suitable for deployment on resource-constrained platforms.
>
> Also, the training time of the OE model is much reduced when compared to the counterpart MoE configuration (***Table 5*** in the main paper; ***Table A9.2.1*** in the supplement file). As we discussed in the Model Complexity section of the paper (***page 9, lines 250-260***), each expert in MoE is trained only on the sub-task specific data whereas the OE is trained on the full training dataset. So, the OE model not only benefits from training on a larger dataset, but also from sub-task specialization via the feature transformations.
>
> [1] Taal, Cees H., et al. "An algorithm for intelligibility prediction of time–frequency weighted noisy speech." IEEE Transactions on audio, speech, and language processing 19.7 (2011): 2125-2136.
>
> [2] T. H. Falk et al., A non-intrusive quality and intelligibility measure of reverberant and dereverberated speech. IEEE Trans. Audio Speech Lang. Process. 2010;18(7):1766–1774.
>
> [3] J. F. Santos and T. H. Falk. Updating the srmr-ci metric for improved intelligibility prediction for cochlear implant users. IEEE Trans. on Audio, Speech, and Lang. Process. 22(12):2197–2206, 2014.
>
> [4] K. Chu et al., Suppressing reverberation in cochlear implant stimulus patterns using time-frequency masks based on phoneme groups. Proc. Meet. Acoust., 50(1), 2022.
>
> [5] He, Shwai, et al., Merging Experts into One: Improving Computational Efficiency of Mixture of Experts. In Proc. Conf. Empir. Methods Nat. Lang. Process. (EMNLP), 2023, pages 14685–14691.

---

> ### Author Response · Authors · 2025-08-06
>
> Dear reviewer 5JAh,
>
> We are following up to get clarity on your acknowledgement. We would appreciate a response on whether our rebuttal addressed your questions, particularly relating to the STOI metric and prior work on merging of experts, to make a final recommendation.
>
> Thanks.

---

> > ### Comment · Area_Chair_Htpo · 2025-08-07
> >
> > Dear Reviewer 5JAh,
> >
> > the authors have provided a rebuttal to your review. Has the rebuttal addressed your concerns? Do you have any remaining questions for the authors?
> >
> > Kind regards, AC

---

### Official Review · Reviewer_q66g · 2025-07-01

**Clarity:** 3
**Significance:** 3
**Originality:** 2
**Rating:** 4
**Confidence:** 4

**Summary:**

The paper proposes a speech dereverberation framework targeting Cochlear Implant(CI) device. It attempts to improve upon an existing MoE approach by proposing a single expert model and a phoneme-based feature transformation block. The feature transformation block learns a scale-shift operation to transform the input speech features based on the predicted phoneme label and the expert model works on these features to estimate the masks for speech dereverberation task.

**Questions:**

I have concerns about the proposed scale-shift feature transformation. As I expected, the shift operation would try to learn shift vectors for each phoneme so that the transformed feature clusters are pushed far apart and it gets easier for the model to reduce the training loss (Fig. 5). I suspect this will lead to poor generalization as the clusters grow apart with enough training. Was there any constraint/regularization on the shift vectors? Would it also help to use some kind of contrastive/regularization loss to keep the clusters close by and to maintain continuity in the feature space?

Is the phoneme classifier well trained? Why is the phoneme classification accuracy (Table 1) so low? In the HINT dataset, for some RIRs, accuracy is lower than 30% (whereas the phoneme /sil/ itself is around 31% of the test data).

Is the Fig A6.1.1correct? Are the shift and scale images swapped? It’s counter intuitive to me  that shift is not able to change the cluster locations but scale alone can.

**Ethical Concerns:**

["NO or VERY MINOR ethics concerns only"]

**Final Justification:**

The paper has some flaws but overall a reasonably good submission, especially due to it's thorough evaluation.

Adding this comment on Aug 12:
The issue is that paper takes on a very niche/narrow application and the contributions are not revealing new insights into machine learning, deep learning, or audio. That said, this narrow application is an important one, and the paper has systematically done a suite of thorough experiments. I am truly on the fence about this paper and wouldn't object if the paper gets rejected.

**Limitations:**

Yes.

**Quality:**

2

**Strengths And Weaknesses:**

Strengths:
The paper is well motivated and easy to follow.
The paper targets a very important application of Cochlear Implant for hearing impaired individuals.
Substantial memory and computational complexity reduction compared to baselines.
Ablation experiments are performed to establish understanding of the proposed method.

Major Weaknesses:
Performance improvement is minor from provided baselines specially in more realistic noisy conditions. The much simpler PI model outperforms the proposed method in most of the SNR conditions (Table A7.1).
Limited performance comparisons. Comparisons with existing dereverberation baselines such as [1], [2] not provided.
Since the model is built for practical purposes,I feel a user subjective quality test under real conditions is necessary.


[1] Zhao et al.  A deep learning based segregation algorithm to increase speech intelligibility for hearing-impaired listeners in reverberant-noisy conditions. J Acoust Soc Am. 2018 Sep
[2] Healy et al. A deep learning algorithm to increase intelligibility for hearing-impaired listeners in the presence of a competing talker and reverberation. J Acoust Soc Am. 2019 Mar

Minor weaknesses:
No demo samples are provided. In the absence of subjective quality tests from human users, I would have appreciated some samples of the model's produced output and baselines to better judge the performance.

---

> ### Author Rebuttal · Authors · 2025-07-31
>
> Thank you for spending your time to review our paper and providing valuable feedback. Please find below responses to your comments.
>
> Our work on speech dereverberation is *significant for individuals with hearing impairments* as there is currently no solution in cochlear implants (and hearing aids) that directly addresses reverberation (***lines 50-55***). Even in the absence of noise, individuals with auditory prostheses often struggle to understand speech in reverberation [1], which can negatively impact their quality of life. For example, the learning experience of individuals with hearing impairments [1, 2], which typically involves single talker scenarios in a classroom/lecture hall. We will discuss the impact of reverberation in the revised paper.
> [1] P. Zahorik, "Spatial hearing in rooms and effects of reverberation," in Binaural Hearing, Cham: Springer Int. Publ., 2021, pp. 243–280.
> [2] F. Iglehart, "Speech perception in classroom acoustics by children with cochlear implants and with typical hearing," Am. J. Audiol., 2016;25(2):100-9.
> # C1: Performance in Noisy-reverberant Conditions
> - While our work focused on dereverberation, we tested models in noisy reverberant conditions to evaluate generalizability to unseen conditions. We applied the models *trained only on reverberant speech* on noisy reverberant conditions, creating a train/test data mismatch. In ***Figure A7.1*** (Appendix), the PI model outperforms the OE particularly at lower SNRs. Nonetheless, the OE with known phonemes provides the best overall performance, indicating that the limiting factor is the phoneme classifier.
> - The OE model in the paper used a LSTM for the phoneme classifier. We updated the phoneme classifier with GRU+Attention (GRU+A) network (***Section A.9 and Table A9.1.1***, Supplementary file). New results in Table R1 below show improvements with OE with the new phoneme classifier, outperforming the PI model in reverberant-only (***Table R1 (a)***) and a wider range of SNR levels in noisy-reverberant conditions (***Table R1(b)***).
>
> *Table R1. Mean (± 95% confidence interval) of objective speech intelligibility scores for reverberant (Rev) or noisy Rev signal, direct path (DP) signal and mask estimation methods: phoneme independent (PI), phoneme-based mask estimated by OE with ideal phoneme knowledge ($OE^k$) and phoneme probabilities ($OE^p$) from LSTM (PC-1) and GRU+A (PC-2) classifiers. Bold indicates the highest value among non-oracle models.*
>
> **(a)  Reverberant-only conditions**
> ||$Rev$|$PI$|$OE^p$ (PC-1)|$OE^p$ (PC-2)|$OE^k$|
> | ----------- | ------------- | ------------- | ------------- | ------------- | ------------- |
> |SRMR-CI|1.302$\pm$0.007|1.733$\pm$0.009|1.794$\pm$0.010|$\textbf1.874\pm0.010$|1.938 $\pm$0.009|
> |STOI|0.719$\pm$0.001|0.797$\pm$0.001|0.807$\pm$0.001|$\textbf0.816\pm0.001$|0.836$\pm$0.001|
>
> **(b)	Noisy Reverberant conditions**
> | | -5 | 0 | 5| 10 | 15 | 20 | No noise |
> | ----------- | ------------- | ------------- | ------------- | ------------- | ------------- | ------------- | ------------- |
> | | | | |***SRMR-CI***| | | |
> |Noisy Rev|1.007$\pm$0.015|1.060$\pm$0.014|1.153$\pm$0.015|1.239$\pm$0.017|1.290$\pm$0.018|1.313$\pm$0.018|1.327$\pm$0.018|
> |$PI$|$\textbf1.329\pm0.022$|$\textbf1.431\pm0.022$|$\textbf1.555\pm0.023$|1.649$\pm$0.024|1.702$\pm$0.024|1.725$\pm$0.024|1.812$\pm$0.025|
> |$OE^p$ (PC-1)|1.295$\pm$0.021|1.387$\pm$0.020|1.522$\pm$0.021|1.634$\pm$0.023|1.701$\pm$0.024|1.734$\pm$0.025|1.848$\pm$0.025|
> |$OE^p$(PC-2)|$\textbf1.328\pm0.023$|$\textbf1.428\pm0.021$|$\textbf1.577\pm0.023$|$\textbf1.699\pm0.025$|$\textbf1.769\pm0.026$|$\textbf1.804\pm0.026$|$\textbf1.934\pm0.027$|
> |$OE^k$|1.434$\pm$0.021|1.547$\pm$0.021|1.686$\pm$0.022|1.790$\pm$0.023|1.850$\pm$0.024|1.881$\pm$0.024|2.011$\pm$0.024|
> | | | | |***STOI***| | | |
> |Noisy Rev|0.533$\pm$0.007|0.600$\pm$0.006|0.657$\pm$0.005|0.697$\pm$0.004|0.721$\pm$0.004|0.732$\pm$0.003|0.738$\pm$0.003|
> |$PI$|0.531$\pm$0.009|0.614$\pm$0.008|0.685$\pm$0.006|0.734$\pm$0.005|0.763$\pm$0.004|0.778$\pm$0.003|0.801$\pm$0.003|
> |$OE^p$ (PC-1)|0.546$\pm$0.009|0.629$\pm$0.008|0.700$\pm$0.006|0.750$\pm$0.004|0.779$\pm$0.003|0.793$\pm$0.003|0.812$\pm$0.003|
> |$OE^p$ (PC-2)|$\textbf0.552\pm0.009$|$\textbf0.636\pm0.008$|$\textbf0.708\pm0.006$|$\textbf0.758\pm0.005$|$\textbf0.787\pm0.004$|$\textbf0.801\pm0.003$|$\textbf0.819\pm0.003$|
> |$OE^k$|0.620$\pm$0.007|0.687$\pm$0.006|0.744$\pm$0.005|0.784$\pm$0.004|0.808$\pm$0.003|0.819$\pm$0.002|0.840$\pm$0.002|
> - We included results using GRU+A for PI and OE (mask estimation and phoneme classification) models in the supplement, which show significantly higher performance for OE in reverberation and noisy-reverberant conditions vs. the PI model (***Figures A9.2.3, A9.2.4, A9.3.2, and Table A9.3.1***, Supplementary file).
> We will consolidate the results in the revised paper.
>
> # C2: Comparison with Prior Work
> > **Comparisons with existing dereverberation baselines such as [1], [2] not provided.**
>
> Thanks for bringing [1-2] to our attention. Our goal is to demonstrate that the OE achieves at least equivalent performance to MoE with reduced model complexity. We present key differences with [1-2] based on our objective within the context of real-time processing in low-resource settings. Both studies do speech enhancement in the acoustic-to-acoustic domain, applicable to hearing aids and as a pre-processing step in cochlear implants (CIs). In our work, speech enhancement for CIs occurs within the CI sound processing pipeline (i.e., acoustic-to-electric stimuli) to reduce delay.
>
> *[1] Zhao et al. A deep learning-based segregation algorithm to increase speech intelligibility for hearing-impaired listeners in reverberant-noisy conditions. J Acoust Soc Am. 2018*
>
> - The work uses the same *synthesized* room impulse responses (RIRs) generated in the same simulated room for training and testing. Our work uses **recorded RIRs** from diverse real-world rooms (***Table A1*** in Appendix of our paper) from different dataset for training and testing (**Section 4.2 in our paper**).
> - Different partitions from the *same speech* dataset with the same speaker were used for training and testing, whereas our training and testing use different speech datasets with no overlap in speech material (**Section 4.2 in our paper**).
> - The DNN uses a context window of 9 frames on each side of the current frame, thus *non-causal*, which introduces delay and infeasible for real-time applications. Our approach is *causal* as it relies only on features from the *current* time frame, thus real-time feasible. The paper lacks details about the DNN architecture.
>
> *[2] Healy et al. A deep learning algorithm to increase intelligibility for hearing-impaired listeners in the presence of a competing talker and reverberation. J Acoust Soc Am. 2019*
> - Similar to [1], there is overlap of train and test material: synthesized RIRs generated from the same simulated room, different partitions of the same speech dataset with the same speaker.
> - The DNN is a bidirectional LSTM network, so non-causal.
>
> Thus, a direct comparison of our work with [1-2] is not possible.
> # C3: Sample Speech Files
> We agree on the need for a subjective test to further validate results. We will include CI vocoded speech files in the supplement file of the revised paper. Note that CI vocoded speech sounds very different from natural speech.
>
> # Q1: Constraint/Regularization on Feature Transformations
> We appreciate the reviewer’s observation about feature transformations. To clarify, the size of clusters and distance between clusters with t-SNE are not necessarily informative about the relationship between clusters.
> We apply feature transformations only in the input feature space and the OE training loss is based on the difference between the signal after applying estimated and ideal masks (***equation 1***, main text). Since transformed features are passed through a single expert network and optimized end-to-end with respect to signal loss, the OE balances cluster separation with speech enhancement performance.
>
> We did not apply explicit regularization or contrastive loss on shift or scale parameters. Contrastive and regularized learning may offer potential benefits and will include as future work in the revision.
> # Q2: Phoneme Classifier Accuracy
> Phoneme classifier accuracy is low, particularly for HINT dataset, for these reasons:
> - For real-time speech enhancement, the phoneme classifier is intentionally *lightweight*. Features are extracted at the CI sound processing time frame of 8ms. Phonemes typically range from 70-200 ms [1], making phoneme classification based on an 8ms frame a relatively hard task. Even at low accuracies, noisy phoneme probabilities are still useful to the OE model. Phonemes with similar time-frequency characteristics are likely to be confused with each other (***Figure A4.2***, Appendix). The soft weighting of phoneme-specific masks reduces the impact of phoneme misclassifications. We also show benefits with improved phoneme classification (response C1 above).
> - The silence class includes silent regions at the start and end of the speech file, and silent gaps between phonemes. CUNY datasets have long periods of silence at the start and end of the speech file, which are easy to detect (***Figure A4.2***, Appendix). In contrast, the HINT dataset has minimal silent regions at the start and end of speech (***Figure 2*** in main paper), so the silence class is mostly silent gaps between phonemes, which are harder to detect when filled with reverberant reflections (***Figure A4.2***, Appendix).
> [1] Fant, Gunnar. Speech acoustics and phonetics. Dordrecht: Springer Netherlands, 2005.
>
> # Q3: Figure A6.1.1
> The transformation labels in Figure A6.1.1 are correct. Shift transformation changes cluster origin. Scale transformation changes the cluster size; the scaling is non-uniform as each feature has a different scale parameter and a negative scale value yields a reflection.

---

> > ### Comment · Reviewer_q66g · 2025-08-06
> >
> > I would like to thank the authors for a comprehensive rebuttal. Several of my questions have been addressed well but I am still unclear about the cluster separation issue.
> > Regardless, I see that with the GRU based phoneme classifier the performance has improved in noisy conditions and I
> > am happy to upgrade my review to "borderline accept".

---

> ### Author Response · Authors · 2025-08-06
>
> Dear reviewer q66g,
>
> We would appreciate a response on whether our rebuttal addressed your concerns and questions to make a final recommendation.
>
> Thanks.

---

> ### Author Response · Authors · 2025-08-06
>
> >*I would like to thank the authors for a comprehensive rebuttal. Several of my questions have been addressed well but I am still unclear about the cluster separation issue. Regardless, I see that with the GRU based phoneme classifier the performance has improved in noisy conditions and I am happy to upgrade my review to "borderline accept".*
>
> We thank the reviewer for considering our rebuttal and increasing the score.
>
> We would like to add some more information related to the question on feature clusters. See ***lines 102-103*** in the text: ***"...our approach is to encode subtask selection for specialization implicitly in the input feature space.***"
>
> Similar to how position embedding encodes the position of a token in a sequence, subtask-specific feature transformation is a subtask embedding that encodes a specific downstream task (in this case, phoneme-specific mask estimation). The embedding can be predefined or learned, as in our case (based on the downstream task loss).
>
> We hope that this explanation, in addition to our earlier response (***Q1***), clarifies the feature cluster separation.

---

### Official Review · Reviewer_YCmV · 2025-07-10

**Clarity:** 3
**Significance:** 3
**Originality:** 2
**Rating:** 5
**Confidence:** 5

**Summary:**

The authors propose a variant or alternative to mixture of experts (MoEs) but constrain their analysis to dereverbration of audio signals for cochlear implants and the associated compute and memory constraints. The authors address dereverberation by estimating a time frequency mask which then gets multiplied with the spectrogram representation. The authors propose to have task-specific scale and shift operations and a shared backbone model instead of multiple experts. This idea is validated by comparing first a single model against MoE and then against the proposed task-specific feature transformation with a single model. Experiments are thorough and contain intelligibility metrics obtained after synthesis from the cochlear implant output electrode signals.

**Questions:**

Please clarify if the scale matrix A_k is a dense matrix or just a diagonal matrix. Was not fully clear.

Please clarify why you need a network to predict A_k and b_k from a one-hot vector. It seems wasteful. If you have one hot input and an MLP, the MLP is not required. You can just look up the end result because there are anyway only 40 possible vectors. (line 177). A frequently used term for this is "embedding layer".

Figure 2: Why are true labels with probabilities 0<p<1 instead of either 0 or 1?

Confidence intervals in Table3/4 are odd. Why are they symmetric? Bootstrap-based confidence intervals should be shown as separate columns or smaller numbers but not relative with +/-. Further, are the confidence intervals characterizing the test example variance or the variance between multiple repeated training runs?

**Ethical Concerns:**

["NO or VERY MINOR ethics concerns only"]

**Limitations:**

The biggest limitation of this work is the evaluation on such a niche application. The reader can hardly infer if this Omni-Expert approach applies to their domains and can replace their expensive MoE.

It is also highly likely that the single model capacity and consequentially the MoE model capacity is just much too high for this application. Thus, compressing it into only task-specific feature transformation works. Imagine a more complex task like language modeling where really huge MoEs are used. Given your experimental results, would you advise them to exchange their MoEs with Omni-Expert?

**Paper Formatting Concerns:**

Please check how you write the scale and shift terms. Sometimes they are capital letters, sometimes not.

Text in figures (e.g. Figure 3) borderline too small.

**Quality:**

3

**Strengths And Weaknesses:**

The paper title suggests an alternative to MoEs widely applicable and studied on various domains. However, the abstract then disappoints a little by reducing the scope to audio, to be precise, dereverberation, to be precise, under limited compute, to be precise, on cochlear implants. The contribution is meaningful nonetheless but the title should be toned down or contain "dereverberation".

The experimental validation is very well done. Laudable that you use recorded room impulse responses (RIRs). Even more so that you take test RIRs from a different database than the train RIRs. Good experimental study with three different oracle conditions, good use of direct path signal to get, e.g., alignments and oracle. Often enough I had seen papers where people use the source signal as reference instead of direct path signal (and then wonder why timing is not right).

The references for alternatives to MoEs are a bit sparse. I suggest the conditional scale and shift parameters should cite https://arxiv.org/abs/1707.00683, in particular conditional batch normalization in Section 3. Also https://www.ecva.net/papers/eccv_2022/papers_ECCV/papers/136870299.pdf is a paper in which the backbone is shared and then only task specific (output) branches are used to make this behave like MoE.

---

> ### Author Rebuttal · Authors · 2025-07-30
>
> Thank  you for spending your time reviewing our paper and providing valuable feedback. We are particularly encouraged by the reviewer’s commendation of the key strengths of our work that we were indeed most excited about in our paper.
>
> # Comments:
>
> > **C1: The paper title suggests an alternative to MoEs widely applicable and studied on various domains. However, the abstract then disappoints a little by reducing the scope to audio, to be precise, dereverberation, to be precise, under limited compute, to be precise, on cochlear implants. The contribution is meaningful nonetheless but the title should be toned down or contain "dereverberation".**
>
> We thank the reviewer for highlighting the meaningful contribution of our work. The NeurIPS conference covers a diversity of research topics, and we submitted our paper under the primary area of *Applications*. We believe that researchers from other fields could benefit from learning about practical considerations when deploying algorithms for end-users in real-world conditions. In this case, the current paradigm of scaling with a MoE is not practical given constraints imposed by real-time processing and limited compute-resources. We note that real-time speech enhancement is applicable to cochlear implants (acoustic-to-electric stimuli) and hearing aids (acoustic-to-acoustic), with the former representing the harder case. While our work focuses on speech dereverberation, the core architectural idea of substituting multi-expert inference with subtask-specific transformations applied to a single expert is in principle domain-agnostic. As researchers have developed innovative ways to extend the MoE technique since the original paper, we believe other researchers could advance or adapt the OE technique if it is a suitable alternative.  We will clarify this in the revised paper.
>
> > **C2: The reader can hardly infer if this Omni-Expert approach applies to their domains and can replace their expensive MoE.**
>
> We broadly discussed the societal impact of our work in the *Discussion* section (**lines 284-289**) and will revise for better clarity. As we do not necessarily have expertise in other domains, our intent is not to indicate a universal replacement of MoE but offer guidance to determine if the Omni-Expert approach is a suitable alternative. MoE have been applied at the embedding, token/sequence or task level. Our main goal here is estimating a phoneme-specific mask based on the phoneme in the current time frame and phoneme features exhibit inherent structure and homogeneity; hence the subtasks are defined/known. Similarly, in domains like large language models (LLMs) or computer vision, there is often latent structure among tokens, tasks, or embeddings (e.g., syntactic roles, semantic types, or class labels). These can be used to define subtask-specific transformations or input conditioning, which suggests that our Omni-Expert approach could be applicable. Other variants of the OE may be needed when specialized subtasks are not as well-defined.
>
> > **C3: Imagine a more complex task like language modeling where really huge MoEs are used. Given your experimental results, would you advise them to exchange their MoEs with Omni-Expert?**
>
> In more complex domains, we postulate that our OE approach could offer value where expert specialization is correlated with known input attributes (e.g., task tokens, domain IDs, or modality types). We do not claim OE would match the performance of large-scale MoEs in language modeling, but it may serve as a scalable, cost-effective solution in specific parts of a larger system or in resource-constrained deployments of LLMs.
>
> >**C4: The references for alternatives to MoEs are a bit sparse. I suggest the conditional scale and shift parameters should cite ..., in particular conditional batch normalization in Section 3. Also ... is a paper in which the backbone is shared and then only task specific (output) branches are used to make this behave like MoE.**
>
> We thank the reviewer for highlighting these relevant works and we will reference them in our revised paper. Please note that we removed the links in accordance with the current policy of not including links in the rebuttal response.
>
> *[1] De Vries, Harm, et al. "Modulating early visual processing by language." Advances in neural information processing systems 30 (2017).*
>
> In Conditional Batch Normalization (CBN) [1], a pretrained convolutional neural network is implemented with the idea of predicting additive adjustments to the batch normalization scaling and shifting parameters based on an external conditioning vector (e.g., a language embedding). Instead of directly predicting the full parameters, CBN predicts small residual changes to pretrained batch normalization parameters, which helps stabilize training and preserve pretrained knowledge. This approach maintains pretrained performance and allows scalable adaptation with minimal additional parameters.
>
> Thus, our proposed method shares the concept of conditional scaling and shifting with the following differences:
>
> •	Instead of modulating batch normalization parameters in each block of the residual network, we apply conditional scale and shift transformations directly to the input features before passing them to a single expert network.
>
> •	Unlike CBN’s residual updates to frozen parameters of a pretrained residual network, our transformations are learned jointly end-to-end with the rest of the expert network.
>
> *[2] Xu, Xiaogang, et al. "Mtformer: Multi-task learning via transformer and cross-task reasoning." European Conference on Computer Vision. Cham: Springer Nature Switzerland, 2022.*
>
> MTFormer framework [2] employs a transformer-only multi-task learning (MTL) architecture consisting of a shared feature extractor (encoder and decoder) and lightweight task-specific branches. MTFormer uses self-task attention within each task-specific branch and introduces cross-task attention modules to enable feature propagation among tasks, achieving efficient parameter sharing and superior performance on multi-task problems.
>
> •	While MTFormer separates specialization through task-specific output branches after a shared transformer backbone, our Omni-Expert model realizes expert specialization within a single expert network by applying task-specific feature transformations. This design avoids multiple output heads and keeps the model size minimal.
>
> > **C5: Please check how you write the scale and shift terms. Sometimes they are capital letters, sometimes not.**
>
> We thank the reviewer for noticing the inconsistency. Our intention was to use capital letters for the general case of the linear transformation matrix ($\mathbf{A}_k$) and lowercase notation for the specific case of scale transformation ($\textit{\textbf{a}} _k$). The shift transformation has the same dimension as the feature space; we will correct the typo in line 113 to ($\mathbf{b}_k$). For clarity and consistency, we will revise Equation (3) and the surrounding text to align with this convention and make the transition between the general and specific notations more explicit (see continuation of explanation in later section).
>
> > **C6: Text in figures (e.g. Figure 3) borderline too small.**
>
> We will increase font size in the revised manuscript.
>
> # Questions:
>
> >**Q1: Please clarify if the scale matrix A_k is a dense matrix or just a diagonal matrix. Was not fully clear.**
>
> We appreciate the question regarding the structure of linear transformation $\mathbf{A}_k$ and complete our explanation from the previous section. $\mathbf{A}_k \in \mathbb{R}^n$ ($n$ is the length of the feature vector and different linear transformations (scale, shear, reflection, etc.) have different matrix forms; a scale transformation is a diagonal matrix, which simplifies to element-wise multiplication. To reduce the number of $\mathbf{A}_k$ parameters that need to be estimated, we use a scale transformation.
>
> >**Q2: Please clarify why you need a network to predict A_k and b_k from a one-hot vector. It seems wasteful. If you have one hot input and an MLP, the MLP is not required. You can just look up the end result because there are anyway only 40 possible vectors. (line 177). A frequently used term for this is "embedding layer".**
>
> We appreciate the reviewer’s insightful comment. Since the mapping from phoneme label to phoneme-specific transformation is fixed, we use MLPs only during training to learn the feature transformation parameters. We precompute and store the final transformation vectors for all 40 phonemes, effectively reducing the process during inference to a lookup table. While we discuss this in the *Model Complexity* section of our paper (***Section 5.5, lines 257-260 and Table 5***), we recognize the need for a different placement of this information and a better description of the model architecture (including figure) and training for better clarity. We will revise the description of the OE model architecture and training in Section 3 of the paper.
>
> >**Q3: Figure 2: Why are true labels with probabilities 0<p<1 instead of either 0 or 1?**
>
> Figure 2 shows soft posterior probability outputs from the phoneme classifier, rather than hard classifier decisions.
>
> > **Q4: Confidence intervals in Table3/4 are odd. Why are they symmetric? Bootstrap-based confidence intervals should be shown as separate columns or smaller numbers, but not relative with +/-. Further, are the confidence intervals characterizing the test example variance or the variance between multiple repeated training runs?**
>
> The confidence intervals in Table 3/4 are derived from the standard error (hence symmetric) and characterize the variability across the test examples.

---

### Decision · Program_Chairs · 2025-09-17

**Decision:**

Accept (poster)

**Comment:**

The paper proposes an omni-expert (OE) model as a more efficient alternative to mixture-of-experts (MoE) for speech dereverberation in cochlear implant devices. Instead of training multiple experts, the approach uses a single backbone model combined with a task-specific feature transformation block that applies learned scale-and-shift operations based on phoneme information. Experiments show that the OE model achieves better intelligibility and computational efficiency compared to both single-model and MoE baselines.

The paper is well-motivated, easy to follow and targets an important real-world application (cochlear implants for hearing-impaired users). Methodology-wise, it yields a significant reduction in memory and computational complexity compared to MoE. The experimental evaluation is thorough. However, the title and scope are sold broader than the actual contribution and the paper misses some related work. Also the improvements to the baselines are rather slim at times.

All reviewers were quite positive about the paper. In the rebuttal and the discussion phase, this opinion was confirmed.